# Differential contributions of synaptic and intrinsic inhibitory currents to speech segmentation via flexible phase-locking in neural oscillators

**Benjamin R. Pittman-Polletta**[1]*, **Yangyang Wang**[2], **David A. Stanley**[1], **Charles E. Schroeder**[3], **Miles A. Whittington**[4], **Nancy J. Kopell**[1]

**1** Department of Mathematics & Statistics, Boston University, Boston, Massachusetts, United States of America, **2** Department of Mathematics, University of Iowa, Iowa City, Iowa, United States of America, **3** Translational Neuroscience Division, Center for Biomedical Imaging and Neuromodulation, Nathan S. Kline Institute for Psychiatric Research, Orangeburg, New York, United States of America, **4** Hull York Medical School, York, United Kingdom

* benpolletta@gmail.com

**Data Availability Statement:** The code for the paper is available at https://github.com/benpolletta/flexible-oscillator-segmentation.

## Abstract

Current hypotheses suggest that speech segmentation—the initial division and grouping of the speech stream into candidate phrases, syllables, and phonemes for further linguistic processing—is executed by a hierarchy of oscillators in auditory cortex. Theta (~3-12 Hz) rhythms play a key role by phase-locking to recurring acoustic features marking syllable boundaries. Reliable synchronization to quasi-rhythmic inputs, whose variable frequency can dip below cortical theta frequencies (down to ~1 Hz), requires "flexible" theta oscillators whose underlying neuronal mechanisms remain unknown. Using biophysical computational models, we found that the flexibility of phase-locking in neural oscillators depended on the types of hyperpolarizing currents that paced them. Simulated cortical theta oscillators flexibly phase-locked to slow inputs when these inputs caused both (i) spiking and (ii) the subsequent buildup of outward current sufficient to delay further spiking until the next input. The greatest flexibility in phase-locking arose from a synergistic interaction between intrinsic currents that was not replicated by synaptic currents at similar timescales. Flexibility in phase-locking enabled improved entrainment to speech input, optimal at mid-vocalic channels, which in turn supported syllabic-timescale segmentation through identification of vocalic nuclei. Our results suggest that synaptic and intrinsic inhibition contribute to frequency-restricted and -flexible phase-locking in neural oscillators, respectively. Their differential deployment may enable neural oscillators to play diverse roles, from reliable internal clocking to adaptive segmentation of quasi-regular sensory inputs like speech.

## Author summary

Oscillatory activity in auditory cortex is believed to play an important role in auditory and speech processing. One suggested function of these rhythms is to divide the speech stream

**Funding:** BRPP, DAS, CES, and NJK were supported by National Institutes of Health (nih.gov) grant P50-MH109429. MAW was supported by Wellcome Trust (wellcome.ac.uk) grant #098353. CES was supported by National Institutes of Health grant R01-MH111439. The funders had no role in study design, data collection and analysis, decision to publish, or preparation of the manuscript.

**Competing interests:** The authors have declared that no competing interests exist.

into candidate phonemes, syllables, words, and phrases, to be matched with learned linguistic templates. This requires brain rhythms to flexibly synchronize with regular acoustic features of the speech stream. How neuronal circuits implement this task remains unknown. In this study, we explored the contribution of inhibitory currents to flexible phase-locking in neuronal theta oscillators, believed to perform initial syllabic segmentation. We found that a combination of specific intrinsic inhibitory currents at multiple timescales, present in a large class of cortical neurons, enabled exceptionally flexible phase-locking, which could be used to precisely segment speech by identifying vowels at mid-syllable. This suggests that the cells exhibiting these currents are a key component in the brain's auditory and speech processing architecture.

## 1 Introduction

Conventional models of speech processing [1–3] suggest that decoding proceeds by matching chunks of speech of different durations with stored linguistic memory patterns or templates. Recent oscillation-based models have postulated that this template-matching is facilitated by a preliminary segmentation step [4–8], which determines candidate speech segments for template matching, in the process tracking speech speed and allowing the adjustment (within limits) of sampling and segmentation rates [9, 10]. Segmentation plays a key role in explaining a range of counterintuitive psychophysical data that challenge conventional models of speech perception [8, 11–13], and conceptual hypotheses [6, 7, 14–18] suggest cortical rhythms entrain to regular acoustic features of the speech stream [19–22] to effect this preliminary grouping of auditory input.

Speech is a multiscale phenomenon, but both the amplitude modulation of continuous speech and the motor physiology of the speech apparatus are dominated by syllabic timescales —i.e., $\delta/\theta$ frequencies (∼1-9 Hz) [23–27]. This syllabic timescale information is critical for speech comprehension [11, 12, 26, 28–31], as is speech-brain entrainment at $\delta/\theta$ frequencies [32–38], which may play a causal role in speech perception [39–42]. Cortical $\theta$ rhythms—especially prominent in the spontaneous activity of primate auditory cortex [43]—seem to perform an essential function in syllable segmentation [11–13, 37], and seminal phenomenological [11] and computational [44–47] models have proposed a framework in which putative syllables segmented by $\theta$ oscillators drive speech sampling and encoding by $\gamma$ (∼30-60 Hz) oscillatory circuits. The fact that oscillator-based syllable boundary detection performs better than classical algorithms [45, 46] argues for the role of endogenous rhythmicity—as opposed to merely event-related responses to rhythmic inputs—in speech segmentation and perception.

However, there are issues with existing models. *In vitro* results show that the dynamics of cortical, as opposed to hippocampal [48], $\theta$ oscillators depend on intrinsic currents at least as much as (and arguably more than) synaptic currents [49, 50]. Yet existing models of oscillatory syllable segmentation assume $\theta$ rhythms are paced by synaptic inhibition [45, 47], and employ methodologies—integrate-and-fire neurons [45] or one-dimensional oscillators [47]—incapable of capturing the dynamics of intrinsic currents. This is important because the variability of syllable lengths between syllables, speakers, and languages, as well as across linguistic contexts, demands "flexibility"—the ability to phase-lock, cycle-by-cycle, to quasi-rhythmic inputs with a broad range of instantaneous frequencies [6, 12], including those below an oscillator's intrinsic frequency—of any cortical $\theta$ oscillator tasked with syllabic segmentation. In contrast to this functional constraint, (synaptic) inhibition-based rhythms have been shown to exhibit *inflexibility* in phase-locking, especially to input frequencies lower than their intrinsic frequency [51,

52]. Furthermore, the pattern of spiking exhibited by a flexible $\theta$ rhythm—which we show depends markedly on the intrinsic currents it exhibits—has important implications for downstream speech processing, being hypothesized to determine how and at what speed $\beta$- ($\sim$15-30 Hz) and $\gamma$-rhythmic cortical circuits sample and predict acoustic information [47, 53]. And while much is known about phase-locking in neural oscillators [54–58], the existing literature sheds little light on these issues: few studies have examined the physiologically relevant "strong forcing regime", in which input pulses are strong enough to elicit spiking [59]; little work has explored how oscillator parameters influence phase-locking to inputs much slower or faster than an oscillator's intrinsic frequency [60]; and few published studies explore oscillators exhibiting intrinsic outward currents on multiple timescales [61].

In addition, syllable boundaries lack reliable acoustic markers, and the consonantal clusters that mark *linguistic* syllable boundaries have higher information density than the high energy and long-duration vowels at their center. This has led to the suggestion that reliable speech-brain entrainment may reverse the syllabic convention, relying on the high energy vocalic nuclei at the center of each syllable to mark segmental boundaries [16] and enable both robust determination of these boundaries and dependable sampling of the consonantal evidence that informs segment identity. These reversed "theta-syllables" are hypothesized to be the candidate cortical segments distinguished and passed downstream for further processing [16] by auditory cortical $\theta$ rhythms, but whether $\theta$ rhythms differentially entrain to different speech channels (associated with the acoustics of consonants and vowels) remains unexamined, as does the impact of such differential entrainment on syllabic timescale speech segmentation.

Motivated by these issues, we explored whether and how the biophysical mechanisms giving rise to cortical $\theta$ oscillations affect their ability to flexibly phase-lock to inputs containing frequencies slower than their intrinsic frequency. We tested the phase-locking capabilities of biophysical computational models of neural $\theta$ oscillators, parameterized to spike intrinsically at 7 Hz, and containing all feasible combinations of: (i) $\theta$-timescale subthreshold oscillations (STOs) resulting from an intrinsic $\theta$-timescale hyperpolarizing current (as observed in $\theta$-rhythmic layer 5 pyramids [50, 62], and whose presence is denoted by "M" in the name of the model); (ii) an intrinsic "super-slow" ($\delta$-timescale) hyperpolarizing current (also observed *in vitro* [50], and present in models with an "S"); and (iii) $\theta$-timescale synaptic inhibition, as previously modeled [45] (present in models with an "I"). We drove these oscillators with synthetic periodic and quasi-periodic inputs, as well as speech inputs derived from the TIMIT corpus [63]. To determine whether and how these oscillators' spiking activity could contribute to meaningful syllabic-timescale segmentation, we used speech-driven model spiking to derive putative segmental boundaries, and compared these boundaries' temporal and phonemic distribution to syllabic midpoints obtained from phonemic transcriptions.

Models exhibiting the combination of STOs and super-slow rhythms observed *in vitro* (models MS and MIS) showed markedly more flexible phase-locking to synthetic inputs than primarily inhibition-paced models (models I, MI, and IS), and yielded segmental boundaries closer to syllabic midpoints, even when phase-locking to speech was hampered by a higher overall level of inhibition (model MIS). Exploring the activation of these three inhibitory currents immediately prior to spiking revealed that flexible phase-locking was driven by a novel complex interaction between $\theta$-timescale STOs and super-slow K currents. This interaction enabled a buildup of outward (inhibitory) current during input pulses, sufficiently long-lasting to silence spiking during the period between successive inputs, even when this period lasted for many $\theta$ cycles, that was absent from oscillators paced by synaptic inhibition. All our models phase-locked most strongly to mid-vocalic channels and produced segmental boundaries predominately during vocalic phonemes, supporting the notion that $\theta$-rhythmic syllable segmentation may make use of $\theta$-syllables rather than conventional, linguistically defined ones.

## 2 Results

### 2.1 Modeling cortical $\theta$ oscillators

To investigate how frequency flexibility in phase-locking depends on the biophysics and dynamics of inhibitory currents, we employed Hodgkin-Huxley type computational models of cortical $\theta$ oscillators (Fig 1). In these models, $\theta$ rhythmicity was paced by either or both of two mechanisms: synaptic inhibition with a fast rise time and a slow decay time as in the hippocampus [48] and previous models of syllable segmentation [45]; and $\theta$-frequency sub-threshold oscillations (STOs) resulting from the interaction of a pair of intrinsic currents activated at sub-threshold membrane potentials—a depolarizing persistent sodium current and a hyperpolarizing and slowly activating m-current [49]. A super-slow potassium current introduced a $\delta$ timescale into the dynamics of some models and helped to recreate dynamics observed *in vitro* [50]. Thus, in addition to spiking and leak currents, our models included up to three types of outward—i.e. hyperpolarizing and thus spike suppressing, and here termed inhibitory—currents: an m-current or slow potassium current ($I_\mathrm{m}$) with a voltage-dependent time constant of activation of $\sim$10-45 ms; recurrent synaptic inhibition ($I_\mathrm{inh}$) with a decay time of 60 ms; and a super-slow K current ($I_{\mathrm{K_{SS}}}$) with (calcium-dependent) rise and decay times of $\sim$100 and $\sim$500 ms, respectively. The presence of these three hyperpolarizing currents was varied over six models—M, I, MI, MS, IS, and MIS—whose names indicate the presence of each current: M for the m-current, I for synaptic inhibition, and S for the super-slow K current (Fig 1).

We began by qualitatively matching *in vitro* recordings from layer 5 $\theta$-resonant pyramidal cells [50] (Fig 2). As their resting membrane potential is raised over a few mV, these RS cells exhibit a characteristic transition from tonic $\delta$-rhythmic spiking to tonic $\theta$-rhythmic spiking through so-called mixed-mode oscillations (MMOs, here doublets of spikes spaced a $\theta$ period apart occurring at a $\delta$ frequency) [50]. *In vitro* data suggests that this pattern of spiking is independent of recurrent synaptic inhibition, arising instead from intrinsic inhibitory currents. To replicate this behavior, we constructed a Hodgkin-Huxley neuron model paced by both $I_\mathrm{m}$ and $I_{\mathrm{K_{SS}}}$ (Figs 1F and 2A). While *in vitro*, these layer 5 $\theta$-rhythmic pyramidal cells receive $\delta$-rhythmic EPSPs, this rhythmic excitation is not required in our model, which exhibited MMOs in response to tonic input (Fig 2D).

We then constructed five additional models based on model MS (Fig 1). To compare the performance of this model to inhibition-based oscillators, we obtained model IS by replacing $I_\mathrm{m}$ with feedback synaptic inhibition $I_\mathrm{inh}$ from a SOM-like interneuron (Fig 1D), adjusting the leak current and the conductance of synaptic inhibition to get a frequency-current (FI) curve having a rheobase and inflection point similar to that of model MS (Fig 1D). In the remaining models, only the leak current conductance was changed, to enable 7 Hz tonic spiking at roughly similar values of $I_\mathrm{app}$; except for the presence or absence of the three inhibitory currents, all other conductances were identical to those in models MS and IS (see Methods). Two models without $I_{\mathrm{K_{SS}}}$ (model M and model I, Fig 1A and 1C) were constructed to explore this current's contribution to model phase-locking. Two more models were constructed with both $I_\mathrm{m}$ and $I_\mathrm{inh}$ to explore the interactions of these currents (Fig 1B and 1E). (Models with neither $I_\mathrm{m}$ nor $I_\mathrm{inh}$ lacked robust 7 Hz spiking). For all simulations, we chose and fixed $I_\mathrm{app}$ so that all models exhibited intrinsic rhythmicity at the same frequency, 7 Hz (Fig 1, small red circles), allowing us to directly compare the frequency range of phase-locking between models.

### 2.2 Phase-locking to strong forcing by simulated inputs

We tested the entrainment of these model oscillators using simulated inputs strong enough to cause spiking with each input "pulse".

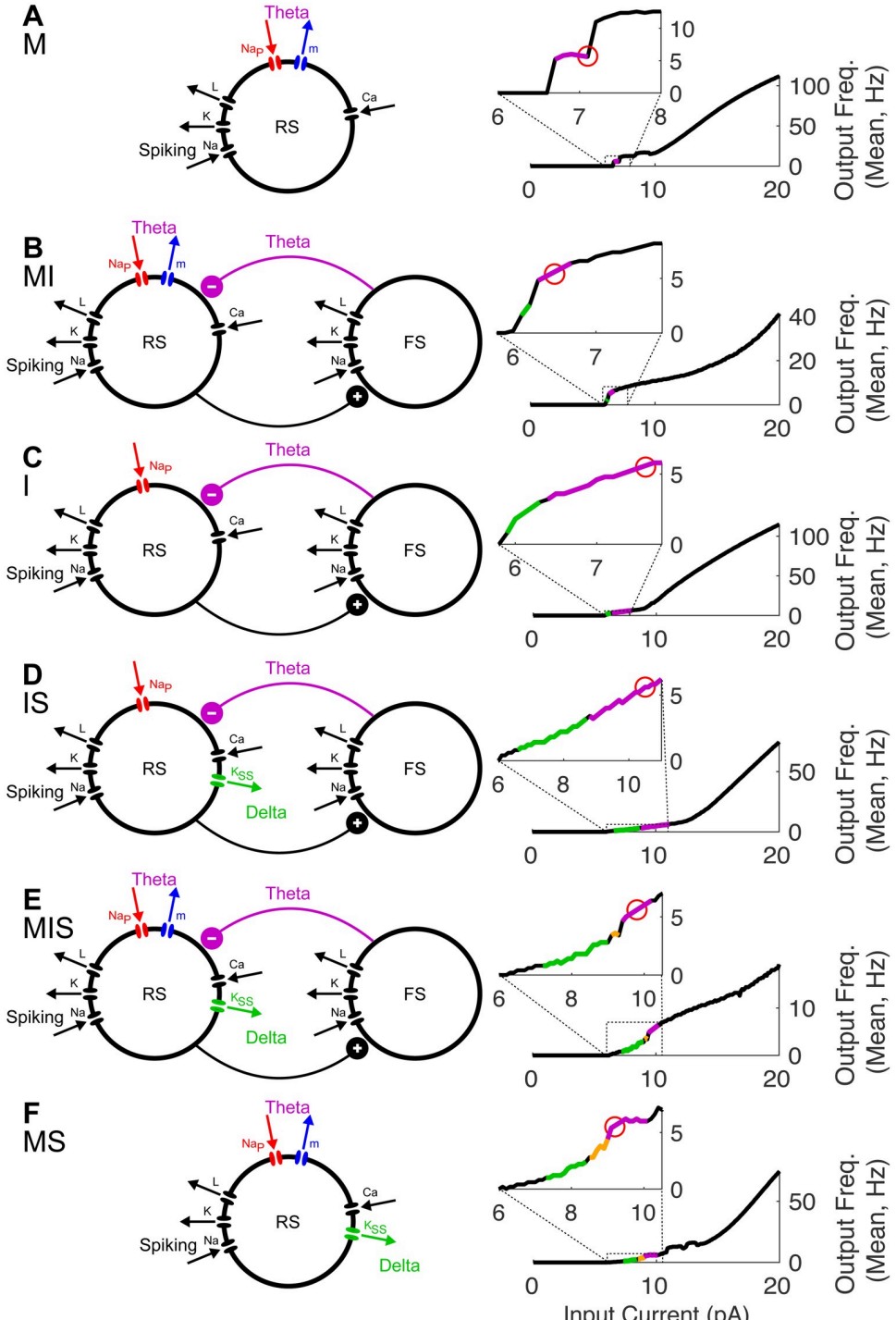

**Fig 1. Model θ oscillators.** For each model (A-F), schematics (left) show the currents present, color-coded according to the timescale inhibition ($\delta$ in green, $\theta$ in purple). FI curves (right) show the transition of spiking rhythmicity through $\delta$ and $\theta$ frequencies as $I_{app}$ increases ($\delta$ in green, $\theta$ in purple, and MMOs in gold); the red circle indicates the point on the FI curve at which $I_{app}$ was fixed, to give a 7 Hz firing rate.

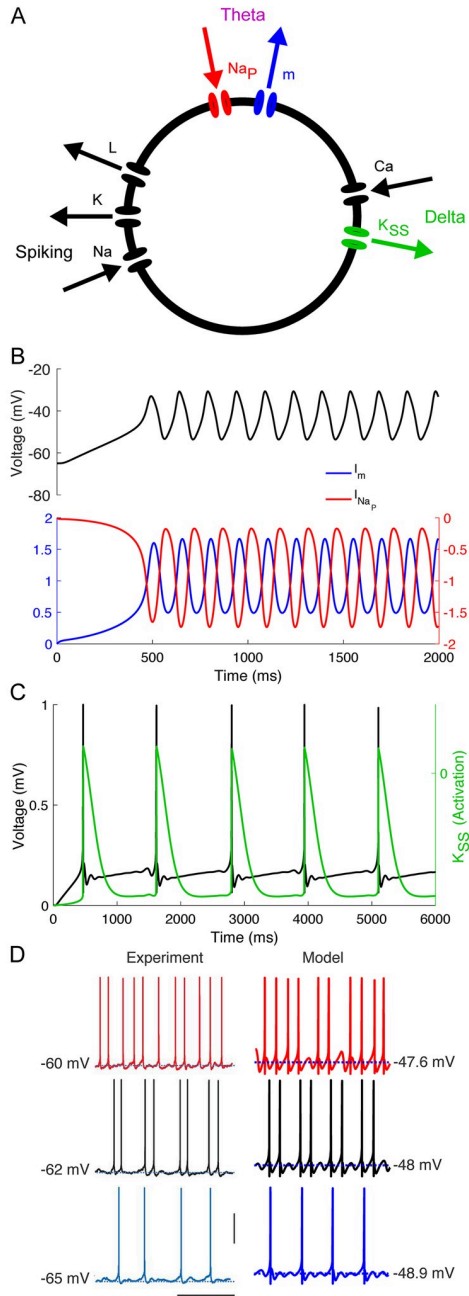

**Fig 2. Model MS reproduces *in vitro* data.** (A) Diagram of model MS. Arrows indicate directions of currents (i.e., inward or outward). (B) $\theta$ timescale STOs arise from interactions between m- and persistent sodium currents in a model without spiking or *Ca*-dependent currents (only $g_\mathrm{m}$ and $g_{\mathrm{Na_P}}$ nonzero). (C) $\delta$ timescale activity-dependent hyperpolarization arises from a super-slow K current. (D) Comparison between *in vitro* (adapted from [50]) and model data (vertical bar 50 $\mu$V, horizontal bar 0.5 ms).

**2.2.1 Rhythmic inputs.**   To begin mapping the frequency range of phase-locking in our models, we measured model phase-locking to regular rhythmic inputs, modeled as smoothed square-wave current injections to the RS cells of all three models. The frequencies of these inputs ranged from 0.25 to 23 Hz, and their duty cycles were held constant at 1/4 of the input period (see Methods), to mimic the bursts of excitation produced by deep intrinsic bursting (IB) cells projecting to deep regular spiking (RS) cells [50]. For inputs at all frequencies, the total (integrated) input over 30 s was normalized, and multiplied by a gain varied from 0 to 4. Entrainment was measured as the phase-locking value (PLV) of RS cell spiking to the input rhythm phase (see Methods, Section 4.3).

The results of these simulations are shown in Fig 3, with models ordered by increasing frequency flexibility of phase-locking, as measured by the lower frequency limit of appreciable phase-locking. The most flexible model (MS) was able to phase lock to input frequencies as low as 1.5 Hz even when input strength was relatively low, while the least flexible model (M) was unable to phase-lock to input frequencies below 7 Hz. For high enough input strength, all models were able to phase-lock adequately to inputs faster than 7 Hz, up to and including the fastest frequency we tested (23 Hz). However, much of this phase-locking occurred with less than one spike per input cycle (see white contours, Fig 3). Notably, models MI and MIS exhibited one-to-one phase-locking to periodic inputs at input strengths twice as high as other models. Simulations showed that this was due to a higher overall level of inhibition, as the range of input strengths over which one-to-one phase-locking was observed increased with the conductances of both $I_{\mathrm{m}}$ and $I_{\mathrm{inh}}$ (S1 Fig).

**2.2.2 Quasi-rhythmic inputs.**   Next, we tested whether the frequency selectivity of phase-locking exhibited for periodic inputs would carry over to quasi-rhythmic inputs, by exploring how model $\theta$ oscillators phase-locked to trains of input pulses in which pulse duration, inter-pulse duration, and pulse waveform varied from pulse to pulse. The latter were chosen (uniformly) randomly from ranges of pulse "frequencies", "duty cycles", pulse shape parameters, and onset times (see Methods, Eq (3)). To create a gradient of sets of (random) inputs with different degrees of regularity, we systematically varied the intervals from which input parameters were chosen (see Methods, Section 4.3.2); we use "bandwidth" here as a shorthand for this multi-dimensional gradient in input regularity. Input pulse trains with a "bandwidth" of 1 Hz were designed to be similar to the 7 Hz periodic pulse trains from Section 2.2.1.

For these "narrowband", highly regular inputs, all six models showed a high degree of phase-locking (Fig 4). In contrast, phase-locking to "broadband" inputs was high only for the models that exhibited broader frequency ranges of phase-locking to regular rhythmic inputs. At high input strengths, model MS in particular showed a high level of phase-locking that was nearly independent of input regularity (Fig 4). Notably, model MIS mirrored the ability of model MS to phase-lock to broadband inputs at high input intensity, while showing frequency selective phase-locking at low input intensity. Indeed, model MIS phase-locked to weak, narrowband quasi-rhythmic inputs better than any other model, perhaps due to its large region of one-to-one phase-locking (Fig 4).

## 2.3 Speech entrainment and segmentation

**2.3.1 Phase-locking to speech inputs.**   We then tested whether frequency flexibility in response to rhythmic and quasi-rhythmic inputs would translate to an advantage in phase-locking to real speech inputs selected from the TIMIT corpus [63]. We also tested how phase-locking to the speech amplitude envelope might differ between auditory frequency bands, examining the response of each model to 16 different auditory channels, ranging in frequency from 0.1 to 3.3 kHz, extracted by a model of the cochlea and subcortical nuclei responsible for

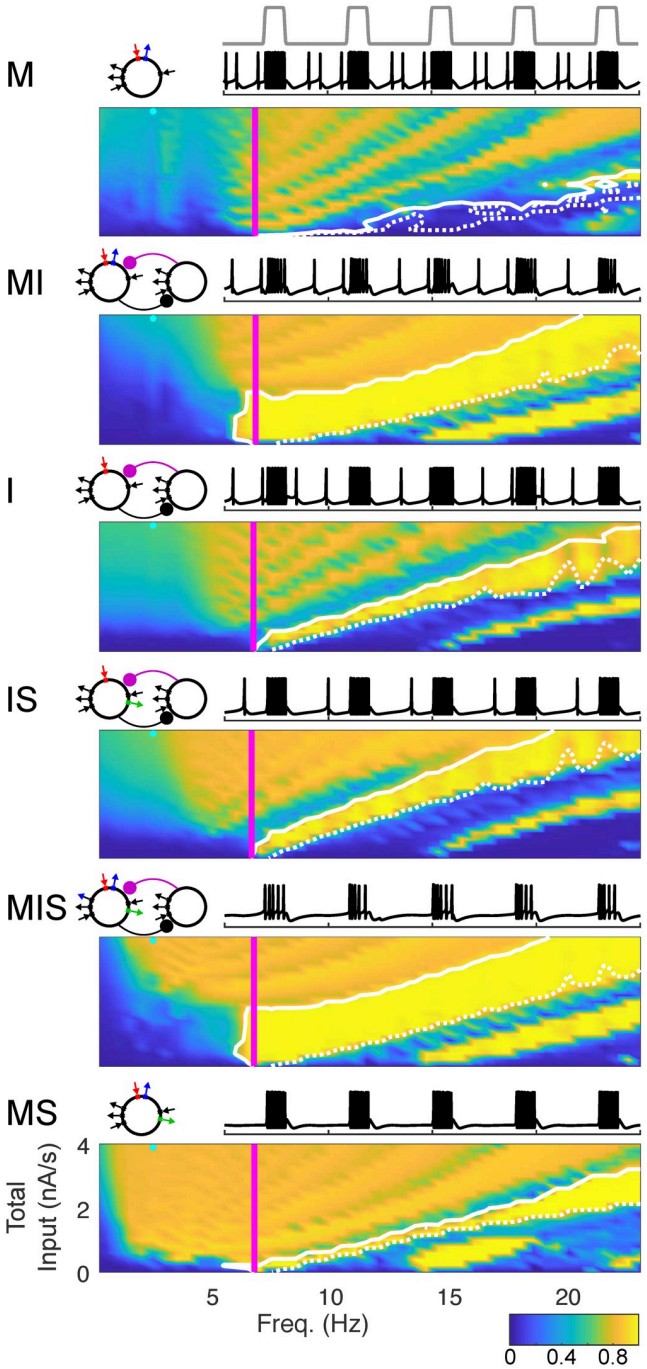

**Fig 3. Phase-Locking as a Function of Periodic Input Frequency & Strength.** False-color images show the (spike-rate adjusted) phase-locking value (PLV, see Section 4.3) of spiking to input waveform. Vertical magenta lines indicating intrinsic spiking frequency. Solid white contour indicates boundary of phase-locking with one spike per cycle; dotted white contour indicates boundary of phase-locking with 0.9 spikes per cycle. Bands in false-color images of PLV are related to the number of spikes generated per input cycle: the highest PLV occurs when an oscillator produces one spike per input cycle, and PLV decreases slightly (from band to band) as both the strength of the input and the number of spikes per input cycle increases. Schematics of each model appear above and to the left; sample traces of each model appear above and to the right (voltage traces in black, input profile in gray, two seconds shown, input frequency 2.5 Hz, total input −3.4 nA/s, as indicated by cyan dot on the false-color image). Total input per second was calculated by integrating input over the entire simulation.

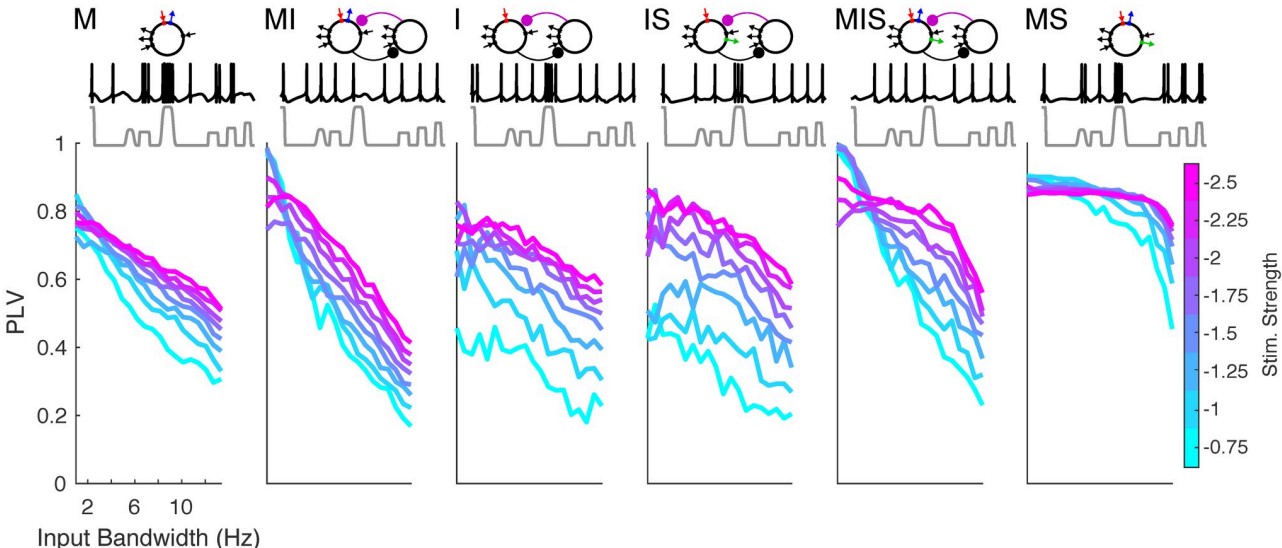

**Fig 4. Phase-locking to quasi-rhythmic inputs.** Plots show the (spike-rate adjusted) phase-locking value of spiking to input waveform, for inputs of varying input strength as well as varying bandwidth and regularity (see Section 4.3.2). All inputs have a center frequency of 7 Hz. Schematics of each model appear above. Sample traces from each model are shown in black, in response to inputs shown in gray, having a bandwidth of 10.65 Hz and an input gain of 1.1; 1.1 second total is shown.

auditory processing [64] from 20 different sentences selected blindly from the TIMIT corpus. We varied the input strength of these speech stimuli with a multiplicative gain that varied between 0 and 2, and assessed the PLV of RS cell spiking to auditory channel phase (Fig 5). All models exhibited a linear increase in PLV with input gain, and the strongest phase-locking to the mid-vocalic channels ($\sim$0.206-0.411 kHz, with peak phase-locking to 0.357 kHz; $p < 10^{-10}$, S2 Fig). To compare the models' performance without the heterogeneous contribution of sub-optimal channels and gains, we ran further simulations with 1000 sentences using only the highest level of multiplicative gain (2) and the 0.233 kHz channel (shown to be optimal among a larger number of channels run in the course of our segmentation simulations, see Section 2.3.2 below). For these simulations, comparisons between models showed that the strength of phase-locking was consistent with the models' ability to phase-lock flexibly to periodic and varied pulse inputs, with the notable exception that models MIS and MI exhibited the weakest performance (S2 Fig). We hypothesized this was again due to their high level of inhibition.

**2.3.2 Speech segmentation by phase-locked cortical $\theta$ oscillators.** We next sought to assess whether phase-locking to speech inputs could contribute to functionally relevant speech segmentation, and whether the validity of this segmentation might differ between auditory frequency bands. To do so, we divided the auditory frequency range into 8 sub-bands consisting of 16 channels each, and drove 16 copies of each of our six models with speech input from each sub-band. We used a simple sum-and-threshold mechanism, intended to approximate the integration of the 16 model oscillators' spiking by a shared postsynaptic target, to translate model activity into syllabic-timescale segmental boundaries (see Methods, Section 4.4.1). We then compared these model-derived segmental boundaries to transcription-derived boundaries, extracted from phonemic transcriptions of the TIMIT corpus (see Methods, Section 4.4.2). Since all our models exhibited the highest levels of phase-locking to the mid-vocalic channels, and since the high energy phase for these channels occurs between syllabic boundaries, we compared model-derived segmental boundaries to the midpoints of transcription-derived syllables, computing a normalized point-process metric $D_{VP,50}$ [65] that penalized

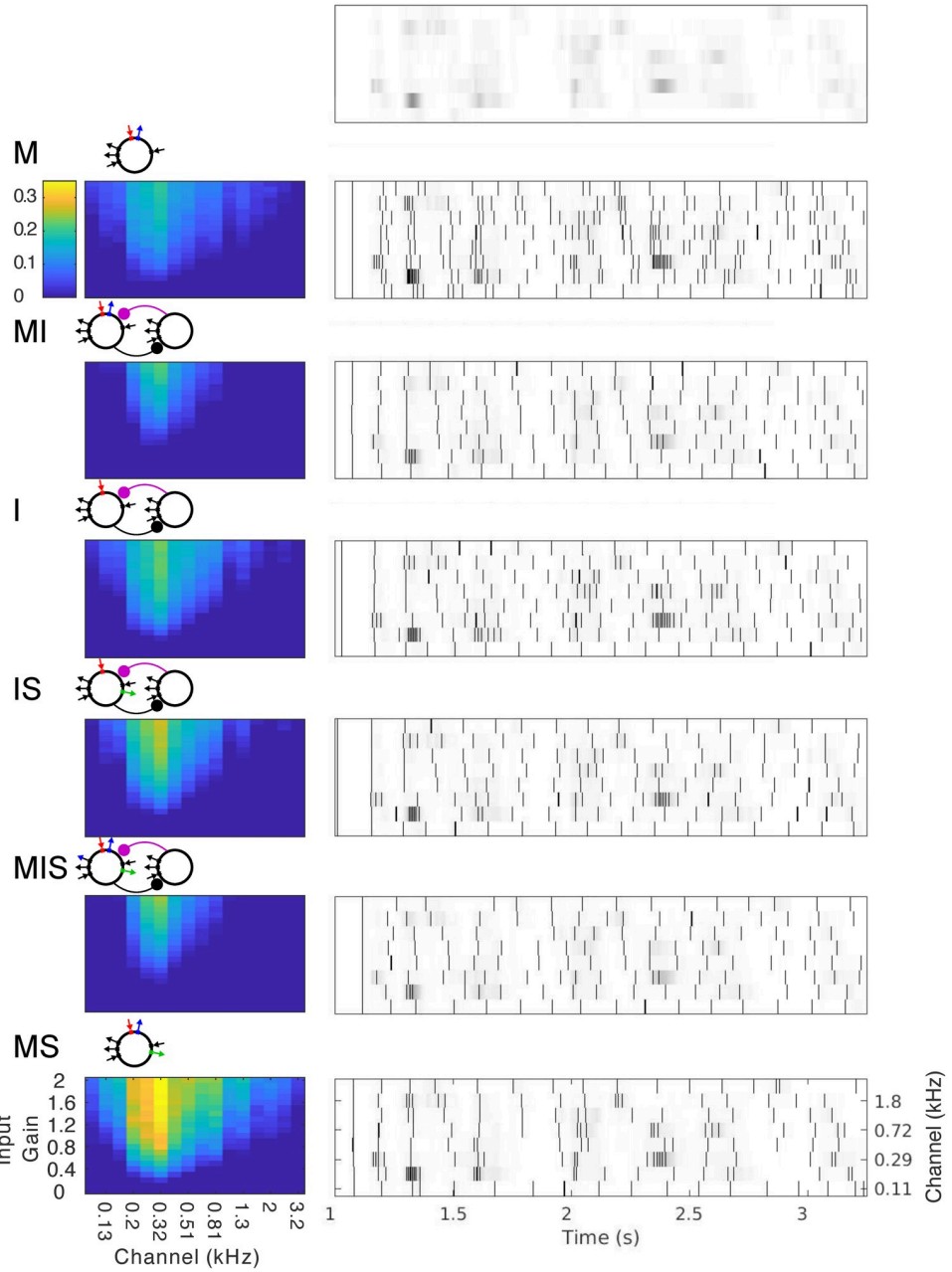

**Fig 5. Phase-locking to speech inputs.** False-color plots (left) show the mean (spike-rate adjusted) PLV of spiking to speech input waveforms, for different auditory channels (x-axis) as well as varying input strengths (y-axis). Gray-scale plots (right) show the spiking response of each model to a selection of 8 auditory channels for a single example sentence. The amplitude of each auditory channel is shown in gray-scale; the top plot shows these amplitudes without any model response. The spiking in response to each channel is overlaid as a raster plot, with a black vertical bar indicating each spike. Schematics of each model appear to the upper left.

model boundaries shifted by more than 50 ms from a syllabic midpoint, as well as "extra" model boundaries and "missed" syllable midpoints (see Methods, Section 4.4.3). Because syllabic midpoints are not necessarily linguistically meaningful, the functional utility of model-derived boundaries may not depend on whether they occur exactly at (or within 50 ms of) mid-syllable. Hypothesizing that model-derived boundaries might function simply to identify

particular phonemes (i.e., vowels), we also examined the phonemic distribution of model-derived boundaries.

The derivation of boundaries from model spiking depended on two parameters—a decay timescale $w_s$ used to sum spikes over time, and a threshold level $r_{thresh}$ used to determine boundary times. In general, the values of the parameters $w_s$ and $r_{thresh}$ dramatically affected segmentation performance (S3 Fig). Intuitively, these parameters may be thought of as analogous to synaptic timescale and efficacy, for example representing maximal NMDA and AMPA conductances, respectively. The ranking of models' segmentation performance depended on the choice of these parameters (S3 Fig), suggesting that a downstream "boundary detector" could "learn" to detect syllable boundaries from the output of the model, by adjusting these parameters.

We thus individually "optimized" each model's performance over a modest set of $w_s$ and $r_{thresh}$ values, finding the $w_s$ and $r_{thresh}$ values for each model that produced the minimum mean $D_{VP,50}$ (for any gain and channel, see Methods, Section 4.4.4). Comparing $D_{VP,50}$ across these "optimized" data sets (S4 Fig) revealed that segmentation performance roughly mirrored entrainment performance, with model MS, the mid-vocalic sub-band (center frequency 0.296 kHz), and the highest gain (2) producing the lowest mean $D_{VP,50}$.

To more rigorously compare model segmentation performance, we ran simulations with 1000 sentences for only the mid-vocalic channel at the highest gain, and once again optimized $w_s$ and $r_{thresh}$ independently for each model (S4 Fig). The resulting ranking across models followed phase-locking flexibility with the exception of model M, which performed as well as model MIS. This tie was surprising, demonstrating the possibility of accurate syllable segmentation even in the absence of high levels of phase-locking to speech inputs. All models, with the exception of model MI, produced a boundary phoneme distribution with a proportion of vowels as high or higher than the proportion of vowels occurring at mid-syllable (Fig 6).

## 2.4 Mechanisms of phase-locking

**2.4.1 Role of post-input spiking delay.** Given that both the most selective and the most flexible oscillator were paced by the m-current, we sought to understand how the dynamics of outward currents contributed to the observed gradient from selective to flexible phase-locking. We hypothesized that phase-locking to input pulse trains in our models depended on the duration of the delay until the next spontaneous spike following a single input pulse. Our rationale was that each input pulse leads to a burst of spiking, which in turn activates the outward currents that pace the models' intrinsic rhythmicity. These inhibitory currents hyperpolarize the models, causing the cessation of spiking for at least a $\theta$ period, and in some cases much longer. If the pause in spiking is sufficiently long to delay further spiking until the next input arrives, phase-locking is achieved, given that the next input pulse will also cause spiking (as a consequence of being in the strong forcing regime). In other words, if $D$ is the delay (in s) between the onset of the input pulse and the first post-input spike, then the lower frequency limit $f^*$ of phase-locking satisfies

$$1/f^* \leq D \Rightarrow f^* \geq 1/D. \tag{1}$$

To test this hypothesis, we measured the delay of model spiking in response to single spike-triggered input pulses, identical to single pulses from the periodic inputs discussed in Section 2.2.1, with durations corresponding to periodic input frequencies of 7 Hz or less, and varied input strengths. The fact that these pulses were triggered by spontaneous rhythmic spiking allowed a comparison between intrinsic spiking and spiking delay post-input (Fig 7A), which showed a correspondence between flexible phase-locking and the duration of spiking delay.

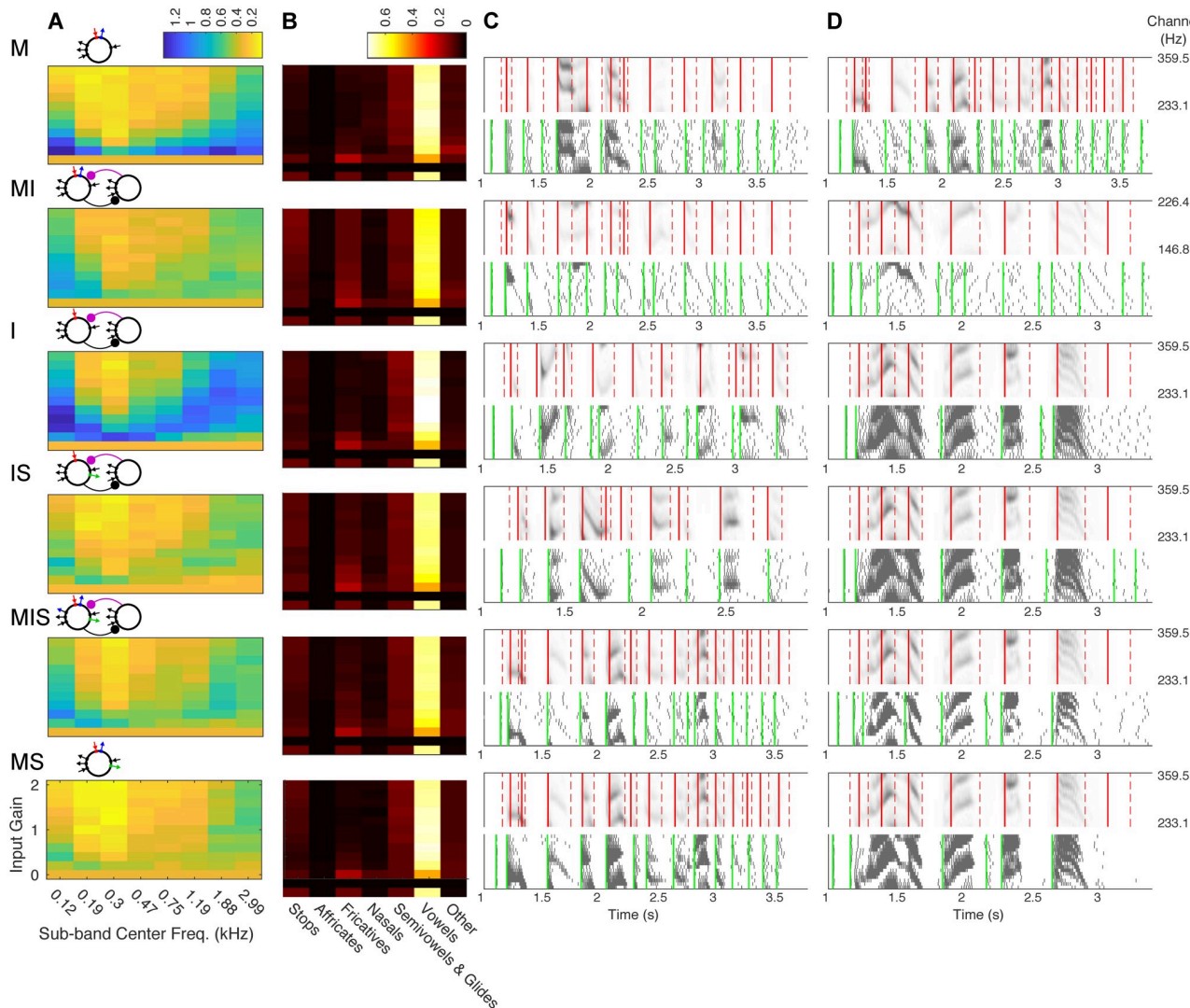

**Fig 6. Speech segmentation.** (A) Mean $D_{VP,50}$ for different auditory sub-bands (x-axis) and varying input strengths (y-axis), for the pair of values taken from $w_s = \{25, 30, \ldots, 75\}$ and $r_{thresh} = \{1/3, .4, .45\ldots, .6, 2/3\}$ that minimized $D_{VP,50}$ for 40 randomly chosen sentences (see Section 4.4.4). Schematics of each model appear to the upper left. (B) The proportion of model-derived boundaries intersecting each phoneme class (x-axis), for the mid-vocalic sub-band (center freq. ∼0.3 kHz) and varying input strengths (y-axis). For comparison, the bottom row shows the phoneme distribution of syllable midpoints. Values of $w_s$ and $r_{thresh}$ are the same as in (A). (C) & (D) Example sentences, model responses, and transcription- and model-derived syllable boundaries. For each model, for the sub-band and input strength with the lowest mean $D_{VP,50}$, the sentences with the lowest (C) and highest (D) $D_{VP,50}$ are shown. Each set of two plots shows the speech input (top panel, gray), syllabic boundaries (red dashed lines), and syllable midpoints (red solid lines); as well as the response of the model (bottom, gray) and the model boundaries (green lines).

We also used spiking delay and Eq (1) to estimate the regions of phase-locking for each model oscillator. In agreement with our hypothesis, the delay-estimated PLV closely matched the profiles of frequency flexibility in phase-locking measured in Section 2.2.1 (Fig 7B).

**2.4.2 Dynamics of inhibitory currents.** To understand how the dynamics of intrinsic and synaptic currents determined the length of the post-input pause in spiking, we examined the gating variables of the three outward currents simulated in our models during both spontaneous rhythmicity and following forcing with a single input pulse (Fig 8). Plotting the relationships between these currents during the time step immediately prior to spiking (Fig 9) offered

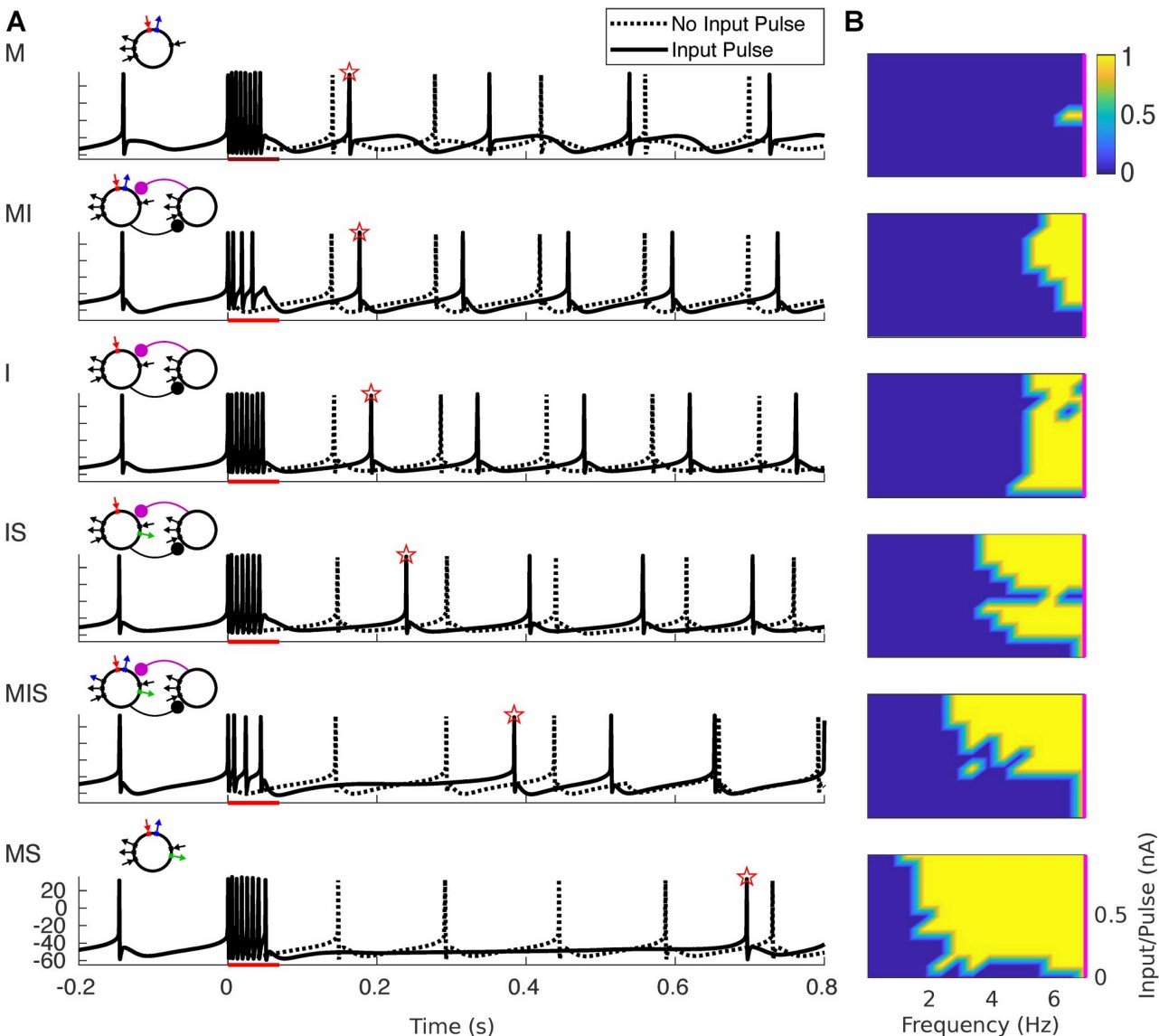

**Fig 7. Delay of spiking in response to single pulse determines phase-locking to slow inputs.** (A) Voltage traces are plotted for simulations both with (solid lines) and without (dotted lines) an input pulse lasting 50 ms. Red bar indicates the timing of the input pulse; red star indicates the first post-input spike. (B) The phase-locking value is estimated from the response to a single input pulse using Eq (1). Frequency was calculated as 1/(4*(pulse duration)), where pulse duration is in seconds. Input per pulse was calculated by integrating pulse magnitude. The magenta line indicates 7 Hz.

insights into the observed gradient of phase-locking frequency flexibility. Below, we describe the dynamics of these outward currents, from simple to complex.

*Synaptic inhibition* Model I spiked whenever the synaptic inhibitory current $I_{inh}$ (Fig 8, purple) or, equivalently, its gating variable, was sufficiently low. This gating variable decayed exponentially from the time of the most recent SOM cell spike; it did not depend on the level of excitation of the RS cell, and thus did not build up during the input pulse. However, post-input spiking delays did occur because RS and SOM cells spiked for the duration of the input pulse, repeatedly resetting the synaptic inhibitory "clock"—the time until $I_{inh}$ had decayed enough for a spontaneous spike to occur. As soon as spiking stopped (at the end of the input

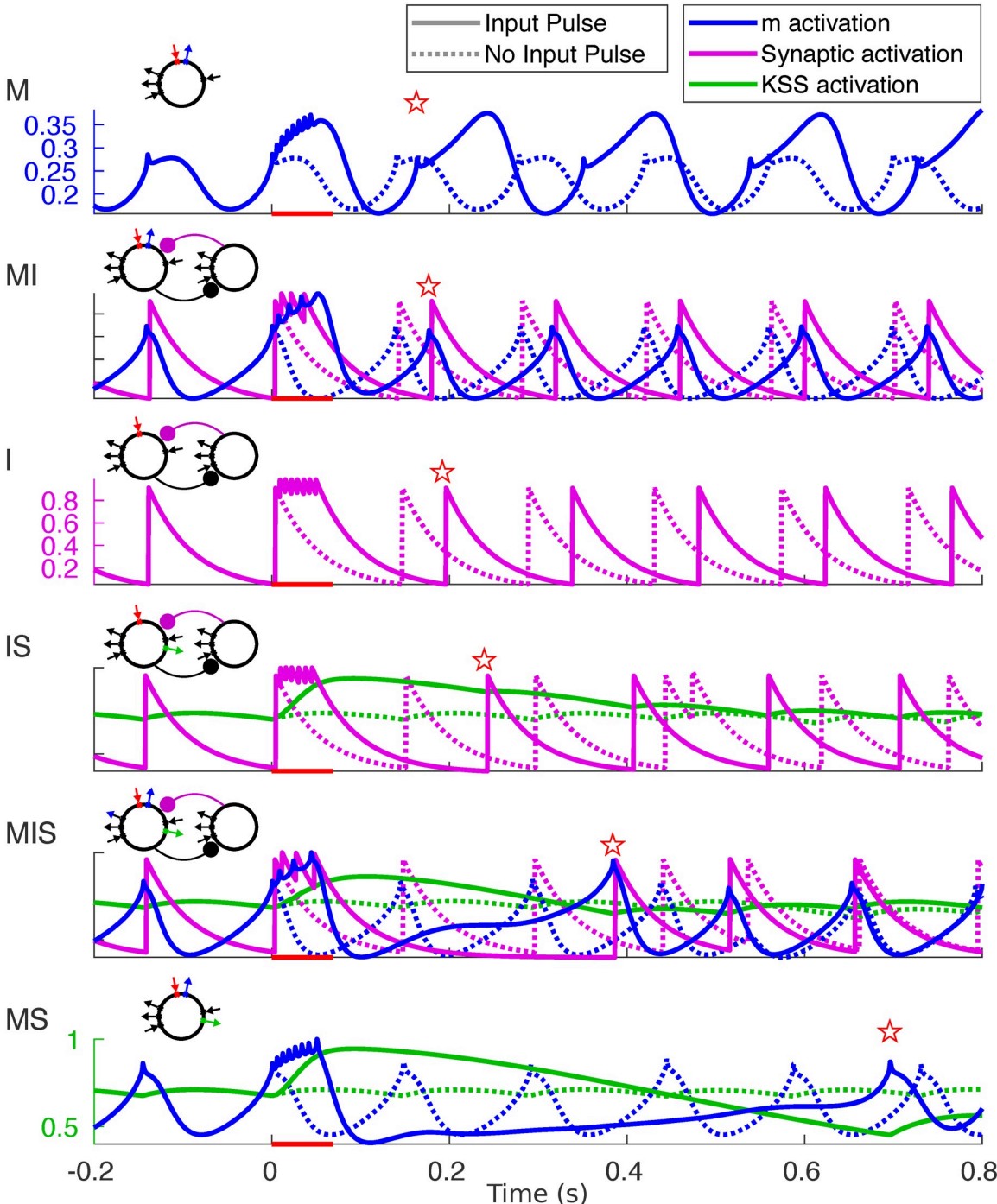

**Fig 8. Buildup of outward currents in response to input pulses.** Activation variables (color) plotted for simulations both with (dotted lines) and without (solid lines) an input pulse lasting 50 ms. Red bar indicates the timing of the input pulse; red star indicates the time of the first post-input spike.

pulse or shortly afterwards—our model SOM interneurons were highly excitable and often exhibited noise-induced spiking after the input pulse), the level of inhibition began to decay, and the next spike occurred one 7 Hz period after the end of the input pulse. For periodic input pulses 1/4 the period of the input rhythm, this suggested that the lower frequency limit $f^*$

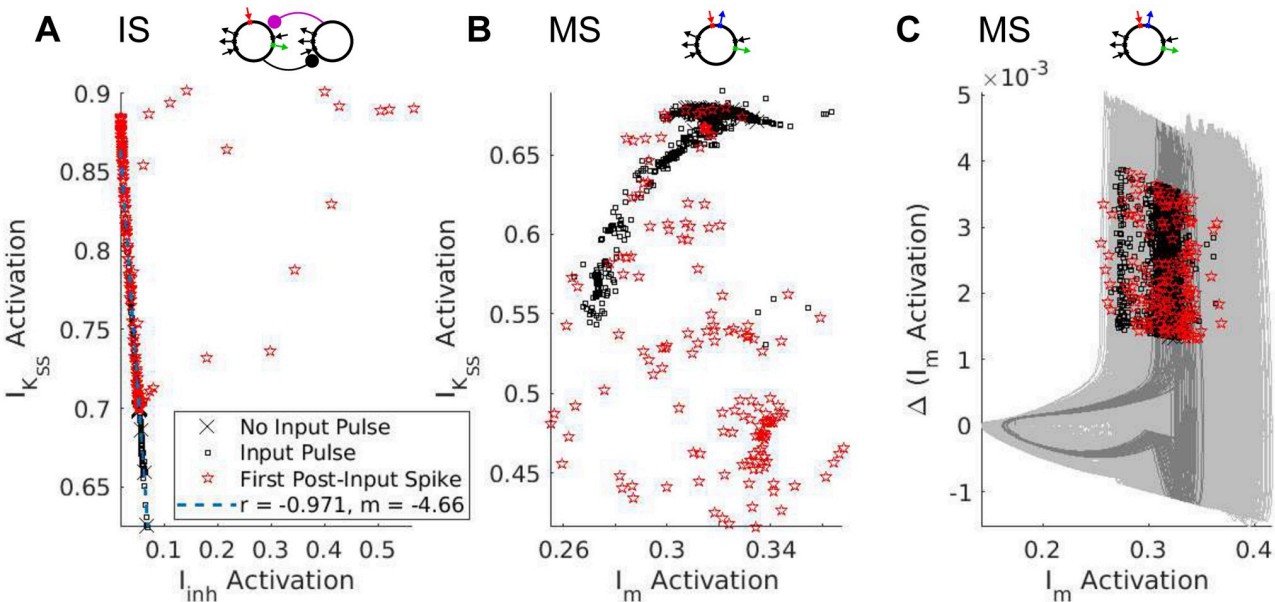

**Fig 9. Linear vs. Synergistic interactions of inhibitory currents.** Plots of the pre-spike gating variables in models IS and MS. (A) The pre-spike activation levels of $I_{inh}$ and $I_{K_{SS}}$ in model IS have a negative linear relationship. (Regression line calculated excluding points with $I_{inh}$ activation > 0.1.) (B) The pre-spike activation levels of $I_m$ and $I_{K_{SS}}$ in model MS do not exhibit a linear relationship. (C) Plotting the activation level of $I_m$ against its first difference reveals that pre-spike activation levels are clustered along a single branch of the oscillator's trajectory. (Light gray curves represent trajectories with an input pulse; dark gray curves represent trajectories without an input pulse).

of phase-locking for model I was determined roughly by the equation

$$D = \frac{1}{4}\left(\frac{1}{f^*}\right) + \frac{1}{7} \geq \frac{1}{f^*} \quad \Rightarrow \quad f^* \geq \frac{21}{4} = 5.25,$$

which corresponded to the limit observed for model I in Figs 3 and 7.

***m-Current*** In contrast, model M did not spike when the m-current gating variable reached its nadir, but during the rising phase of its rhythm (Fig 8). Since the m-current activates slowly, at this phase the upward trajectory in the membrane potential—a delayed effect of the m-current trough—was not yet interrupted by the hyperpolarizing influence of m-current activation. When the cell received an input pulse, the m-current (blue) built up over the course of the input pulse, but since it is a hyperpolarizing current activated by depolarization whose time constant is longest at ~-26 mV and shorter at de- or hyperpolarized membrane potentials, this buildup resulted in the m-current rapidly shutting itself off following the input pulse. This rapid drop resulted in a lower trough, and, subsequently, a higher peak value of the m-current's gating variable (because the persistent sodium current had more time to depolarize the membrane potential before the m-current was activated enough to hyperpolarize it), changing the frequency of subsequent STOs. It didn't, however, affect the model's phase-locking in the strong forcing regime; the fast falling phase of the m-current following the pulse kept the post-input delay small (Fig 8). This "elastic" dynamics offers an explanation for model M's inflexibility: the buildup of m-current during an input pulse leads to a fast hyperpolarization of the membrane potential, which, in turn, causes rapid deactivation of the m-current and subsequent rapid "rebound" of the membrane potential to depolarized levels, preserving the time of the next spike.

***Super-slow K current*** In models with a super-slow K current, this current, like synaptic inhibition, decayed to a nadir before each spike of the intrinsic rhythm. Unlike synaptic

inhibition, $I_{K_{SS}}$ activation built up dramatically during an input pulse (Fig 8, green), and decayed slowly, increasing the latency of the first spike following the input pulse substantially (Fig 7). This slow-building outward current interacted differently, however, with synaptic and intrinsic $\theta$-timescale currents. In model IS, both $I_{inh}$ and $I_{K_{SS}}$ decayed monotonically following an input pulse, until the total level of hyperpolarization was low enough to permit another spike. We hypothesized that $I_{K_{SS}}$ and $I_{inh}$ interacted additively to produce hyperpolarization and a pause in RS cell spiking. In other words, the delay until the next spike was determined by the time it took for a sum of the two currents' gating variables (weighted by their conductances and the driving force of potassium) to drop to a particular level. The fact that we expect this weighted sum of the gating variables to be nearly the same (having value, say, $a^*$) in the time $t^*$ before each spike suggests that the two gating variables are negatively linearly related at spike times:

$$g_{SOM \to RS}s(t^*)(V(t^*) - E_K) + g_{K_{SS}}q(t^*)(V(t^*) - E_K) \simeq a^*$$
$$\Rightarrow \quad q(t^*) \simeq -\frac{g_{SOM \to RS}}{g_{K_{SS}}}s(t^*) + \frac{a^*}{g_{K_{SS}}(V(t^*) - E_K)}.$$

Plotting the activation levels of these two currents in the timestep before each spike against each other confirmed this hypothesis (excluding forced spikes and a handful of outliers, Fig 9A).

The interaction between $I_m$ and $I_{K_{SS}}$ was more complex, as seen in model MS. The pre-spike activation levels of these two currents were not linearly related (Fig 9B). When $I_{K_{SS}}$ built up, it dramatically suppressed the level of the m-current gating variable, biasing the competition between $I_m$ and $I_{Na_p}$ and reducing STO amplitude, and the $I_{K_{SS}}$ activation had to decay to levels much lower than "baseline" before the oscillator would spike again. Indeed, spiking appeared to require m-current activation to return *above* "baseline", and also to be in the rising phase of its oscillatory dynamics. The dependence of spiking on the phase of the m-current activation could be seen by plotting the "phase plane" trajectories of the oscillator—plotting the m activation against its first difference immediately prior to each spike—revealing a branch of the oscillator's periodic trajectory along which pre-spike activation levels were clustered (Fig 9C). Plotting the second difference against the first revealed similar periodic dynamics (S5(A) Fig).

The models containing both synaptic inhibition and m-current exhibited similar dynamics to model MS, with a dependence of spiking on the phase of the rhythm in $I_m$ activation being the clearest pattern observable in the pre-spike activation variables (S5(A) and S5(C) Fig). This suggests that the delay following the input pulse in these models also reflects an influence of $\theta$-timescale STOs, which may exhibit more complex interactions with $I_{inh}$ in model MI, similar qualitatively if not quantitatively to their interactions with $I_{K_{SS}}$ in models MS and MIS.

## 3 Discussion

Our results link the biophysics of cortical oscillators to speech segmentation via flexible phase-locking, suggesting that the intrinsic inhibitory currents observed in cortical $\theta$ oscillators [49, 50] may enable these oscillators to entrain robustly to $\theta$-timescale fluctuations in the speech amplitude envelope, and that this entrainment may provide a substrate for enhanced speech segmentation that reliably identifies mid-syllabic vocalic nuclei. We trace the capacity of cortical $\theta$ oscillators for flexible phase-locking to synergistic interactions between their intrinsic currents, and demonstrate that similar oscillators lacking either of these intrinsic currents show markedly less frequency flexibility in phase-locking, regardless of the presence of $\theta$-

timescale synaptic inhibition. These findings suggest that synaptic and intrinsic inhibition may tune neural oscillators to exhibit different levels of phase-locking flexibility, allowing them to play diverse roles—from reliable internal clocks to flexible parsers of sensory input—that have consequences for neural dynamics, speech perception, and brain function.

## 3.1 Mechanisms of phase-locking

For models containing a variety of intrinsic and synaptic currents, spiking delay following a single input pulse was an important determinant of the lower frequency limit of phase-locking in the strong-forcing regime (Fig 7). A super-slow current, $I_{K_{SS}}$, aided the ability to phase-lock to slow frequencies in our models, by building up over a slow timescale in response to burst spiking during a long and strong input pulse. The presence of the super-slow K current increased the frequency range of phase-locking, with every model containing $I_{K_{SS}}$ able to phase-lock to slower periodic inputs than any model without $I_{K_{SS}}$ (Fig 3). The fixed delay time of synaptic inhibition seemed to stabilize the frequency range of phase-locking, while the voltage-dependent and "elastic" dynamics of the m-current seemed to do the opposite. Specifically, the four models containing $I_{inh}$ exhibited an intermediate frequency range of phase-locking, while both the narrowest and the broadest frequency ranges of phase-locking occurred in the four model $\theta$ oscillators containing $I_{m}$; and the very narrowest and broadest ranges occurred in the two models containing $I_{m}$ and lacking $I_{inh}$ (Fig 3).

Our investigations showed that the flexible phase-locking in models MS and MIS resulted from a synergistic interaction between slow and super-slow K currents, demonstrated here—to our knowledge—for the first time. We conjecture that this synergy depends on the sub-threshold oscillations (STOs) engendered by the slow K current (the m-current) in our models, as was suggested by an analysis of the pre-spike activation levels of the inhibitory currents in models IS and MS. In model IS, there were no STOs, and the interaction between $\theta$-timescale inhibition (which was synaptic) and $I_{K_{SS}}$ was additive, so that spikes occurred whenever the (weighted) sum of these gating variables dropped low enough (Fig 9A). In models MIS and MS, where STOs resulted from interactions between the m-current and the persistent sodium current, spiking depended not only on the level of activation of the m-current, but also on the phase of the endogenous oscillation in m-current activation (Fig 9C).

For all models, the frequency flexibility of phase-locking to periodic inputs translated to the ability to phase-lock to quasi-rhythmic (Fig 4) and speech (Fig 5) inputs. While it is reasonable to hypothesize that this is the result of the mechanism of phase-locking in the regime of strong forcing, it is important to note that imperfect phase-locking in our models resulted not only from "extra" spikes in the absence of input (as predicted by this hypothesis), but also from "missed" spikes in the presence of input (Fig 4). A dynamical understanding of these "missed" spikes may depend on the properties of our oscillators in the weak-forcing regime.

Phase-locking of neural oscillators under weak forcing has been studied extensively [54–58]. In this regime, a neural oscillator stays close to a limit cycle during and after forcing, and as a result the phase of the oscillator is well-defined throughout forcing. Furthermore, the change in phase induced by an input is small (less than a full cycle), can be calculated, and can be plotted as a function of the phase at which the input is applied, resulting in a phase-response curve (PRC). Our results pertain to a dynamical regime in which PRC theory does not apply, since our forcing is strong and long enough that our oscillators complete multiple cycles during the input pulse, and as a result the phase at the end of forcing is not guaranteed to be a function of the phase at which forcing began. Furthermore, in oscillators which contain $I_{K_{SS}}$, the dynamics of this slow current add an additional dimension, which makes it impossible to describe the state of these oscillators in terms of a simple phase variable. Not only the phase of

the oscillator, but also its amplitude (which is impacted by the activation of $I_{K_{SS}}$), determine its dynamics.

Previous work has illuminated many of the dynamical properties of the $\theta$-timescale m-current. The addition of an m-current (or any slow resonating current, such as an h-current or other slow non-inactivating K current) changes a neuron from a Type I to a Type II oscillator [66, 67]. The generation of membrane potential resonance (and subthreshold oscillations) by resonating currents is well-studied [49, 68, 69], and recently it has been shown that the $\theta$-timescale properties of the M-current allow an E-I network subject to $\theta$ forcing to precisely coordinate with external forcing on a $\gamma$ timescale [61]. While STOs play an important role in the behaviors of our model oscillators, subthreshold resonance does not automatically imply suprathreshold resonance or precise response spiking [70]. Thus, our results are not predictable (either *a priori* or *a posteriori*) from known effects of the m-current on neuronal dynamics.

Larger (synaptic) inhibition-paced networks have been studied both computationally and experimentally [52, 71–74], and can exhibit properties distinct from our single (RS) cell inhibition-paced models: computational modeling has shown that the addition of E-E and I-I connectivity in E-I networks can yield frequency flexibility through potentiation of these recurrent connections [72, 74]; and experimental results show that amplitude and instantaneous frequency are related in hippocampal networks, since firing by a larger proportion of excitatory pyramidal cells recruits a larger population of inhibitory interneurons [73], a phenomenon which may enable more frequency flexibility in phase-locking. This raises the question of why the brain would select phase-locking flexibility in single cells vs. networks. One possible answer is energetic efficiency. If flexibility in an inhibition-paced oscillatory network depends on recruiting large numbers of inhibitory interneurons, it may be more efficient to utilize a small number of oscillators, each capable (on its own) of entrainment to quasi-rhythmic inputs containing a large range of instantaneous frequencies.

## 3.2 Functional implications for neuronal entrainment to auditory and speech stimuli

Our focus on the $\theta$ timescale is motivated by results underscoring the prominence of theta rhythms in the spontaneous and stimulus-driven activity of primate auditory cortex [43, 75–77] and by evidence for the (causal [39–42]) role of $\delta/\theta$ frequency speech-brain entrainment in speech perception [32–39, 42]. Our results suggest that the types of inhibitory currents pacing cortical $\theta$ oscillators with an intrinsic frequency of 7 Hz determine these oscillators' ability to phase-lock to the (subcortically processed [64]) amplitude envelopes of continuous speech. While an oscillator with an intrinsic frequency of 3 Hz might do an equally good job of phase-locking to strong inputs with frequencies between 3 and 9 Hz, this does not seem to be the strategy employed by the auditory cortex: the frequencies of (low-frequency) oscillations in primate auditory cortex are ∼1.5 and ∼7 Hz, not 3 Hz [43]; existing experimental [43, 78] and computational [79] evidence suggests that cortical $\delta$ oscillators are unlikely to be driven at $\theta$ frequencies even by strong inputs; and MEG studies show that across individuals, speech comprehension is high when cortical frequencies are the same as, or higher than, speech envelope frequencies, and becomes poorer as this relationship reverses [80].

Another important question raised by our results (and by one of our reviewers) is the following: If flexible entrainment to a (quasi-)periodic input depends on the lengths of the delays induced by the input, why go to the trouble of using an oscillator at all, rather than a cell responding only to sufficiently strong inputs? The major difference between oscillators and non-oscillatory circuits driven by rhythmic inputs is what happens when the inputs cease (or

are masked by noise): while a non-oscillatory circuit lapses into quiescence, an oscillator continues spiking at its endogenous frequency. Thus, oscillatory mechanisms can track the temporal structure of speech through interruptions and omissions in the speech signal [16]. This capability is crucial to the adjustment of speech processing to the speech rate, a phenomenon in which brain oscillations are strongly implicated. While (limited) speeding or slowing of *entire* utterances does not affect their intelligibility, altering context speech rate can change the perception of unaltered target words, even making them disappear [81–86]. In recent MEG experiments, brain oscillations entrained to the rhythm of contextual speech persisted for several cycles after a speech rate change [86], with this sustained rhythmic activity associated with altered perception of vowel duration and word identity following the rate change [86]. Multiple hypothetical mechanisms have been proposed to account for these effects: the syllabic rate (as encoded by the frequency of an entrained $\theta$ rhythm) may determine the sampling rate of phonemic fine structure (as effected by $\gamma$ rhythmic circuits) [6, 53]; predictive processing of speech may use segment duration relative to context speech speed as evidence to evaluate multiple candidate speech interpretations [47, 87]; and oscillatory entrainment to the syllabic rate may time relevant calculations, enabling the optimal balance of speed and accuracy in the passing of linguistic information up the processing hierarchy before the arrival of new input—so-called "chunk-and-pass" processing [88].

Recent experiments shed light on the limits of adaptation to (uniform) speech compression, showing that while cortical speech-brain phase entrainment persisted for syllabic rates as high as 13 Hz (a speed at which speech was not intelligible), $\beta$-rhythmic activity was abnormal in response to this unintelligible compressed speech [89]. This work suggests that the upper syllabic rate limit on speech intelligibility arises not from defective phase-locking, but from inadequate time for mnemonic or other downstream processes between syllables [89]. This agrees with our finding that the upper frequency boundary on phase-locking extends well above the upper syllabic rate boundary on speech intelligibility ($\sim$9 Hz), and is largely determined by input strength. Nonetheless, it is noteworthy that task-related auditory cortical entrainment operates most reliably over the 1-9 Hz (syllabic) ranges [75]. Further exploration of how speech compression affects speech entrainment by neuronal oscillators is called for.

Out of our models, MS came closest to spiking selectively at the peaks of the speech amplitude envelope, yet it did not perform perfectly. This was to be expected for a signal as broadband and irregular as the amplitude envelope of speech, which presents challenges to both entrainment and its measurement (see Section 4.3.3). As we've mentioned, defects in phase-locking were also due to both "missed" cycles and "extra" spikes (Fig 5), whose frequency of occurrence was traded off as tonic excitation to model MS was varied: lower levels of tonic excitation led to more precise phase-locking (i.e., fewer extra spikes) but more missed cycles, while higher levels of tonic excitation led to less precise phase-locking but a lower probability of missed cycles (S6 Fig).

## 3.3 Functional implications for speech segmentation

Multiple theories suggest a functional role for cortical $\theta$ oscillations in segmenting auditory and speech input at the syllabic timescale [6, 11–13, 16, 39, 42, 77, 90–92]. To explore the consequences for syllabic segmentation of the different levels of speech entrainment observed in our oscillators, we implemented a simple method to extract putative segmental boundaries from the spiking of multiple (unconnected) copies of our models. Our results serve to demonstrate that the accuracy with which segmental boundaries can be extracted from the spiking of speech-entrained cortical oscillators depends on the particular biophysics of those oscillators. They suggest that the information in the mid-vocalic channels provides an advantage for

entrainment to speech and for syllabic-timescale segmentation. Finally, they open the door to many new questions about the neuronal bases of speech processing.

Our work points to frequency flexibility, which appears to enable segmentation accuracy even at low levels of entrainment to the speech signal (as can be seen by contrasting the segmentation performance of models MIS and MI), as one of the factors that can impact segmentation accuracy. However, it is clear that other factors also contribute. One likely factor is excitability, a "minor theme" that contributed second-order effects to the behaviors of models MI, MIS, and MS (S1, S2, S4 and S6 Figs). While we tuned our models to exhibit the same (7 Hz) frequency of tonic spiking in the absence of (dynamic) input, and attempted to qualitatively match their F-I curves, our models exhibited clear differences in the number of spikes evoked by inputs of the same strength (Figs 5 and 7). It is likely that this in turn impacted the sum-and-threshold mechanism used to extract syllable boundaries. A highly excitable oscillator may respond to speech input with a surfeit of spiking from which accurate syllable boundaries can be carved by the choice of $w_s$ and $r_{\mathrm{thresh}}$; such a mechanism may account for the unexpectedly accurate segmentation performance of model M. The issue of excitability arises again when inquiring into the advantages mid-vocalic channels offer for speech entrainment and segmentation, as these channels differ not only in their frequency content but also in having higher amplitude than other channels. We have chosen not to normalize speech input beyond the transformations implemented by a model of subcortical auditory processing, but investigating how different types of normalization affect speech entrainment and segmentation could illuminate whether mid-vocalic channels' frequency, amplitude, or both are responsible for the heightened functionality they drive.

There remains much to explore about how segmental boundaries may be derived from the spiking of populations of cortical oscillators. While our implementation was extremely simplistic, omitting heterogeneity in parameters or synaptic or electrical connectivity between oscillators, "optimized" model-derived boundaries arose from a relatively complex integration of the rich temporal dynamics of population activity (Fig 6). This contrasts with the regular and highly synchronous spike volleys characterizing previous models of oscillatory syllable segmentation, in which all $\theta$ oscillators received the same channel-averaged speech input [45]. In our implementation, a boundary is signaled when the activity of the oscillator network passes a given threshold, in agreement with recent results showing that neurons in middle STG, a region of auditory cortex implicated in syllable and word recognition, respond to acoustic onset edges (i.e., peaks in the rate of change of the speech amplitude envelope) [93, 94]. This may explain why segmentation failures occurred when the speech amplitude envelope remained high through an extended time period that included multiple syllabic boundaries (Fig 6D).

One way around this is to combine information across, as well as within, auditory subbands. Our work supports the hypothesis that identification of vocalic nuclei, rather than consonantal clusters, is associated with more precise syllabic-timescale segmentation, but it doesn't preclude the use of information about the timing of consonantal clusters to aid segmentation. Interestingly, different auditory cortical regions entrained to different phases of rhythmic (1.6 Hz) stimuli, with 11-15 kHz regions firing during high-amplitude phases and all other regions firing in antiphase, and this alternating response pattern was suggested to relate to the alternation of vowels and consonants in speech [95]. We suggest that a deeper understanding of the dynamic repertoire afforded by the simple model presented here may provide a foundation for future investigations of more complex (and realistic) networks.

Previous work showed that a synaptic inhibition-paced $\theta$ oscillator was able to predict syllable boundaries "on-line" at least as accurately as state-of-the-art offline syllable detection algorithms [45]. While we have not compared our models directly to these syllable detection

algorithms, we explored the performance of synaptic inhibition-paced $\theta$ oscillators similar to those modeled in previous work. In our hands, models paced even in part by synaptic inhibition performed uniformly worse than comparable models paced by intrinsic currents alone at syllabic-timescale segmentation. However, there exist several differences between previous and current implementations—including input (channel averaged and filtered vs. frequency specific), model complexity (leaky integrate-and-fire vs. Hodgkin-Huxley), temporal dynamics of synaptic inhibition (a longer rise time in earlier models), and parameter optimization—all of which may lead to differences in segmentation performance.

This earlier work positioned syllable segmentation and speech recognition by oscillatory networks within the landscape of syllable detection algorithms arising from the fields of linguistics, engineering, and artificial intelligence [45]. While the current work has focused more on how the biophysical implementations of neuronal oscillators impact speech entrainment and segmentation, an understanding of how differences in segmentation performance and location affect speech recognition is an important direction for future work. It remains unclear whether the explicit representation of segmental boundaries contributes to the effects of speech rate and oscillatory phase on syllable and word recognition [77, 81–86, 90], or to the proposed underlying mechanisms that implicate speech segmentation at the neuronal level [6, 47, 53, 87, 88]. Indeed, whether speech recognition in general requires explicit segmentation or only the entrainment of cortical activity to the speech rhythm remains obscure. Cortical $\theta$ oscillators are embedded in a stimulus-entrainable cortical rhythmic hierarchy [43, 92, 95–97], receiving inputs from deep IB cells embedded in $\delta$-rhythmic circuits [43, 50, 62, 97], and connected via reciprocal excitation to superficial RS cells embedded in $\beta$- and $\gamma$-rhythmic circuits [50, 79]. In the influential TEMPO framework, the $\theta$ oscillator is hypothesized to be driven by $\delta$ circuits, and to drive $\gamma$ circuits, with a linkage between $\theta$ and $\gamma$ frequency adjusting the sampling rate of auditory input to the syllabic rate [6, 53]. It has been hypothesized that cortically-identified syllabic boundaries may reset the activity of $\gamma$-rhythmic circuits responsible for sampling and processing incoming syllables, a reset necessary for accurate syllable recognition [6, 44, 47, 53]. By indicating the completion of the previous syllabic segment, they may also trigger the activity of circuits responsible for updating the linguistic interpretations of previous speech [53]. Not only this reset cue, but also $\theta$-rhythmic drive to $\gamma$-rhythmic circuits, is necessary for accurate syllable decoding within this framework [45]. Recent work with leaky-integrate-and-fire models demonstrates that top-down spectro-temporal predictions can be integrated with theta-gamma coupling, with the latter enabling the temporal alignment of the former to acoustic input [47].

Using the output of our models as an input to syllable recognition circuitry—perhaps via $\gamma$-rhythmic circuits [44, 45, 47]—would enable exploration of whether the differences in segmentation accuracy we uncover are functionally relevant for speech recognition. Comparing syllable recognition when these circuits are driven by model-derived segmental boundaries vs. model spiking may shed light on the necessity of explicit segmental boundary representation for syllable recognition. Such research would also provide an opportunity to test claims that "theta syllables" provide more information for syllabic decoding than conventional syllables [16]. Our results support the hypothesis that cortical $\theta$ oscillators align with speech segments bracketed by vocalic nuclei—so-called "theta syllables"—as opposed to conventional syllables, which defy attempts at a consistent acoustic characterization, but are (usually) bracketed by consonantal clusters [16]. These "theta-syllables" are suggested to have information-theoretic advantages over conventional linguistic syllables: the vocalic nuclei of speech have relatively large amplitudes and durations, making them prominent in noise and reliably identifiable [19]; and windows whose edges align with vocalic nuclei center the diphones that contain the majority of the information for speech decoding, ensuring this information is sampled with

high fidelity. These claims, if they prove to have functional relevance, may illuminate how speech-brain entrainment aids speech comprehension in noisy or otherwise challenging environments [98–100]. Connecting the complex and rich dynamics of networks of biophysically detailed neuronal oscillators to plausible speech recognition circuitry may uncover novel functional and mechanistic factors contributing to speech processing and its dysfunctions [101–105].

### 3.4 Versatility in cortical processing through flexible and restricted entrainment

More broadly, there is evidence that cortical $\theta$ oscillators in multiple brain regions, entrained to distinct features of auditory and speech inputs, may implement a variety of functions in speech processing. Different regions of human superior temporal gyrus (STG) respond differentially to speech acoustics: posterior STG responds to the onset of speech from silence; middle STG responds to acoustic onset edges; and anterior STG responds to ongoing speech [93, 94]. Similarly, bilaterally occuring $\delta/\theta$ speech-brain entrainment may subserve hemispherically distinct but timescale-specific functions, with right-hemispheric phase entrainment [97] encoding acoustic, phonological, and prosodic information [33, 97, 99, 106, 107], and left-hemispheric amplitude entrainment [97] encoding higher-level speech structure [38, 108–110] and top-down predictions [111–113]. Frequency flexibility may shed light on how these multiple $\theta$ oscillations are distinguished, collated, and combined. One tempting hypothesis is that the gradient from flexible to restricted phase-locking corresponds to a gradient from stimulus-entrained to endogenous brain rhythms, with oscillators closer to the sensory periphery exhibiting more flexibility and reverting to intrinsic rhythmicity in the absence of sensory input, enabling them to continue to couple with central oscillators that exhibit less phase-locking flexibility. It is suggestive that the conductance of the m-current, which is key to flexible phase-locking in our models, is altered by acetylcholine, a neuromodulator believed to affect, generally speaking, the balance of dominance between modes of internally and externally generated information [62, 114–116].

Indeed, the potential for flexible entrainment does not seem to be ubiquitous in the brain. Hippocampal $\theta$ rhythm, for example, is robustly periodic, exhibiting relatively small frequency changes with navigation speed [117]. It is suggestive that the mechanisms of hippocampal $\theta$ and the neocortical $\theta$ rhythmicity discussed in this paper are very different: while the former is dominated by synaptic inhibition, resulting from an interaction of synaptic inhibition and the h-current in oriens lacunosum moleculare interneurons [48], the latter is only modified by it [50]. Our results suggest that mechanisms like that of hippocampal $\theta$, far too inflexible to perform the segmentation tasks necessary for speech comprehension, are instead optimized for a different functional role. One possibility is that imposing a more rigid temporal structure on population activity may help to sort "signal" from "noise"—i.e., imposing a strict frequency and phase criterion that inputs must meet to be processed, functioning as a type of internal clock. Another possibility is that more rigidly patterned oscillations result from a tight relationship to motor sampling routines which operate over an inherently more constrained frequency range, as, for example, whisking, sniffing, and running are related to hippocampal $\theta$ [118, 119].

Along these lines, it is intriguing that model MIS exhibits both frequency selectivity in phase-locking at low input strengths, and frequency flexibility in phase-locking at high input strengths (Fig 4). Physiologically, input gain can depend on a variety of factors, including attention, stimulus novelty and salience, and whether the input is within- or cross-modality. A mechanism that allows input gain to determine the degree of phase-locking frequency

flexibility could enable the differential processing of inputs based on these attributes. It is tempting to speculate that such differential entrainment may play a role in both the low levels of speech entrainment of model MIS, and in the model's ability to carry out accurate segmentation in spite of it. Perhaps more trenchantly, the phase-locking properties of our models are themselves modulable, allowing the same neurons to entrain differently to rhythmic inputs depending on the neuromodulatory context.

Although from one perspective model MIS is the most physiologically realistic of our models, as neurons in deep cortical layers are likely to exhibit all three outward currents studied in this paper [50], the minimal impact of synaptic inhibition on these large pyramidal cells suggests that model MS is a functionally accurate representation of the majority (by number) of RS cells in layer 5 [62]. It thus represents the main source of $\theta$ rhythmicity in primary neocortex [62], and a major source of cortico-cortical afferents driving "downstream" processing [120, 121]. Its properties may have strong implications for the biophysical mechanisms used by the brain to adaptively segment and process complex auditory stimuli evolving on multiple timescales, including speech.

# 4 Methods

All simulations were run on the MATLAB-based programming platform DynaSim [122], a framework specifically designed by our lab for efficiently prototyping, running, and analyzing simulations of large systems of coupled ordinary differential equations, enabling in particular evaluation of their dynamics over large regions of parameter space. DynaSim is open-source and all models will be made publicly available using this platform.

## 4.1 Model equations

Our models consisted of at most two cells, a regular spiking (RS) pyramidal cell and an inhibitory interneuron with a timescale of inhibition like that observed in somatostatin-positive interneurons (SOM). Each cell was modeled as a single compartment with Hodgkin-Huxley dynamics. In our RS model, the membrane currents consisted of fast sodium ($I_{Na}$), delayed-rectifier potassium ($I_{K_{DR}}$), leak ($I_{leak}$), slow potassium or m- ($I_m$), and persistent sodium ($I_{Na_p}$) currents taken from a model of a guinea-pig cortical neuron [49], and calcium ($I_{Ca}$) and super-slow potassium ($I_{K_{SS}}$, calcium-activated potassium in this case) currents with dynamics from a hippocampal model [123]. The voltage $V(t)$ was given by the equation

$$C\frac{dV}{dt} = I_{app} - I_{Na} - I_{K_{DR}} - I_{leak} - I_m - I_{Na_p} - I_{Ca} - I_{K_{SS}} - I_{inh}$$

where the capacitance $C = 2.7$ reflected the large size of deep-layer cortical pyramidal cells, and $I_{app}$, the applied current, was given by

$$I_{app}(t) = g_{app}\left[\left(\frac{t}{\tau_{trans}}\chi_{\{t \leq \tau_{trans}\}}(t) + \chi_{\{t > \tau_{trans}\}}(t)\right) + p_{noise}W(t)\right]$$

where $\chi_S(t)$ is the function that is 1 on set S and 0 otherwise, the transition time $\tau_{trans} = 500$ ms, the noise proportion $p_{noise} = 0.25$, and $W(t)$ a white noise process. (The applied current ramps up from zero during the first 500 ms to minimize the transients that result from a step current). For SOM cells, the membrane currents consisted of fast sodium ($I_{Na,SOM}$), delayed-rectifier potassium ($I_{K_{DR},SOM}$), and leak ($I_{leak,SOM}$) currents [124]. The voltage $V(t)$ was given

**Table 1. Currents.**

| | |
|---|---|
| $I_{Na}$ & $I_{Na,SOM}$ | $g_{Na} m_{Na}^3 h(V - E_{Na})$ |
| $I_{K_{DR}}$ & $I_{K_{DR},SOM}$ | $g_{K_{DR}} m_{K_{DR}}^4 (V - E_K)$ |
| $I_{leak}$ & $I_{leak,SOM}$ | $g_{leak}(V - E_{leak})$ |
| $I_m$ | $g_m n(V - E_K)$ |
| $I_{Nap}$ | $g_{Nap} m_{Nap}(V - E_{Nap})$ |
| $I_{Ca}$ | $g_{Ca} s^2 (V - E_{Ca})$ |
| $I_{K_{SS}}$ | $g_{K_{SS}} q(V - E_K)$ |
| $I_{inh}$ & $I_{exc}$ | $g_{pre \to post} s_{pre \to post}(V_{post} - E_{pre \to post})$ |

by the equation

$$C_{SOM} \frac{dV}{dt} = I_{app,SOM} - I_{Na,SOM} - I_{K_{DR},SOM} - I_{leak,SOM} - I_{exc}$$

where $C_{SOM} = 0.9$ and $I_{app,SOM}$, the applied current, is constant in time. The form of each current is given in Table 1; equilibrium voltages are given in Table 2; and conductance values for all six models that are introduced in *Results: Modeling cortical θ oscillators* (see Fig 1) are given in Table 3.

The dynamics of activation variable $x$ (ranging over $h$, $m_{K_{DR}}$, $n$, $m_{Nap}$, $s$, and $q$ in Table 1) were given either in terms of its steady-state value $x_\infty$ and time constant $\tau_x$ by the equation

$$\frac{dx}{dt} = \frac{x_\infty - x}{\tau_x},$$

or in terms of its forward and backward rate functions, $\alpha_x$ and $\beta_x$, by the equation

$$\frac{dx}{dt} = (1 - x)\alpha_x - x\beta_x.$$

Only the expressions for $m_{Na}$ differed slightly:

$$m_{Na}(V) = \alpha_m/(\alpha_m + \beta_m), \qquad m_{Na,SOM}(V) = [1 + \exp((-V - 38)/10)]^{-1}.$$

Steady-state values, time constants, and forward and backward rate functions are given in Table 4. For numerical stability, the backwards and forwards rate constants for $q$ and $s$ were

**Table 2. Equilibrium voltages.**

| | RS | FS |
|---|---|---|
| $E_{Na}$ | 40 | 50 |
| $E_K$ | -80 | -95 |
| $E_{leak}$ | -65 | -70 |
| $E_{Nap}$ | 50 | – |
| $E_{Ca}$ | 120 | – |
| $E_{RS \to SOM}$ | | 0 |
| $E_{SOM \to RS}$ | -95 | |

**Table 3. Maximal conductances.**

| Model | M | MI | I | IS | MIS | MS |
|---|---|---|---|---|---|---|
| $g_{Na}$ | 135 | 135 | 135 | 135 | 135 | 135 |
| $g_{K_{DR}}$ | 54 | 54 | 54 | 54 | 54 | 54 |
| $g_{leak}$ | 0.31 | 0.27 | 0.78 | 0.78 | 0.27 | 0.27 |
| $g_m$ | 1.4472 | 1.4472 | 0 | 0 | 1.4472 | 1.4472 |
| $g_{Nap}$ | 0.4307 | 0.4307 | 0.4307 | 0.4307 | 0.4307 | 0.4307 |
| $g_{Ca}$ | 0.54 | 0.54 | 0.54 | 0.54 | 0.54 | 0.54 |
| $g_{K_{SS}}$ | 0 | 0 | 0 | 0.1512 | 0.1512 | 0.1512 |
| $g_{app}$ | -7.1 | -6.5 | -7.6 | -10.5 | -9.8 | -9.2 |
| $g_{Na,SOM}$ | 0 | 100 | 100 | 100 | 100 | 0 |
| $g_{K_{DR},SOM}$ | 0 | 80 | 80 | 80 | 80 | 0 |
| $g_{leak,SOM}$ | 0 | 0.1 | 0.1 | 0.1 | 0.1 | 0 |
| $I_{app,SOM}$ | 0 | 0.95 | 0.95 | 0.95 | 0.95 | 0 |
| $g_{RS \to SOM}$ | 0 | 0.075 | 0.075 | 0.075 | 0.075 | 0 |
| $g_{SOM \to RS}$ | 0 | 0.15 | 0.15 | 0.15 | 0.15 | 0 |

converted to steady-state values and time constants before integration, using the equations

$$x_\infty = \alpha_x \tau_x, \qquad \tau_x = \left( \alpha_x + \beta_x \right)^{-1}.$$

The dynamics of the synaptic activation variable $s$ were given by the equation

$$\frac{ds}{dt} = -\frac{s}{\tau_D} + \frac{1-s}{\tau_R} \left( 1 + \tanh \left( \frac{V_{pre}}{10} \right) \right),$$

with time constants $\tau_R = 0.25$ ms, $\tau_{D,RS \to SOM} = 2.5$ ms, and $\tau_{D,SOM \to RS} = 50$ ms. The conductance $g_{RS \to SOM}$ was selected to preserve a one-to-one spiking ratio between RS and SOM cells.

## 4.2 F-I curves

For these curves, we varied the level of tonic applied current $I_{app}$ over the range from 0 to 200 Hz, in steps of 1 Hz. We measured the spiking rate for the last 5 seconds of a 6 second simulation, omitting the transient response in the first second. The presence of $\delta$ and $\theta$ rhythmicity or MMOs was assessed using inter-spike interval histograms, and thus differs from the (arrhythmic) spike rate.

**Table 4. Activation variable dynamics.**

| | | |
|---|---|---|
| $h$ | $\alpha_h(V) = 0.07 \exp\left(-(V+30)/20\right)$ | $\beta_h(V) = \left(\exp\left(-V/10\right) + 1\right)^{-1}$ |
| $m_{Na}$ | $\alpha_m(V) = -\frac{V+16}{10(\exp(-(V+16)/10)-1)}$ | $\beta_m(V) = 4 \exp\left(-(V+41)/18\right)$ |
| $m_{K_{DR}}$ | $\alpha_m(V) = -0.01\frac{V+20}{\exp(-(V+20)/10)-1}$ | $\beta_m(V) = 0.125 \exp\left(-(V+30)/80\right)$ |
| $n$ | $n_\infty(V) = \left[1 + \exp\left(-(V+35)/10\right)\right]^{-1}$ | $\tau_n(V) = \frac{1000/(3.3 \ast 3^{(34-22)/10})}{\exp\left(\frac{V+35}{40}\right) + \exp\left(\frac{-(V+35)}{20}\right)}$ |
| $m_{Nap}$ | $m_\infty(V) = \left[1 + \exp\left(-(V+40)/5\right)\right]^{-1}$ | $\tau_m = 5$ |
| $s$ | $\alpha_s(V) = 1.6\left(1 + \exp\left(-0.072(V+65)\right)\right)$ | $\beta_s(V) = 0.02\frac{V+51.1}{\exp\left(\frac{V+51.1}{5}\right)-1}$ |
| $q$ | $\alpha_q(C_{Ca}) = \min\left(0.1 C_{Ca}, 1\right)$ | $\beta_q = 0.002$ |
| $h_{SOM}$ | $h_\infty(V) = \left[1 + \exp\left((V+58.3)/6.7\right)\right]^{-1}$ | $\tau_h(V) = 0.225 + 1.125\left[1 + \exp\left((V+37)/15\right)\right]^{-1}$ |
| $m_{K_{DR},SOM}$ | $m_\infty(V) = \left[1 + \exp\left((-V-27)/11.5\right)\right]^{-1}$ | $\tau_m(V) = 0.25 + 4.35\left[1 + \exp\left(-|V+10|/10\right)\right]^{-1}.$ |

## 4.3 Phase-locking to rhythmic, quasi-rhythmic, and speech inputs

In addition to the tonic applied current $I_{app}$, to measure phase-locking to rhythmic, quasi-rhythmic, and speech inputs, we introduced time-varying applied currents. These consisted of either periodic pulses ($I_{PP}$), variable-duration pulse trains with varied inter-pulse intervals ($I_{VP}$), or speech inputs ($I_{speech}$).

The (spike rate adjusted) phase-locking value (PLV, [125]) of the oscillator to these inputs was calculated with the expressions

$$PLV = \left(n_s |MRV|^2 - 1\right)/(n_s - 1), \qquad MRV = \frac{1}{n_s}\sum_{i=1}^{n_s} \exp\left(\sqrt{-1}\phi_I(t_i^s)\right),$$

where MRV stands for mean resultant vector, $n_s$ is the number of spikes, $t_i^s$ is the time of the $i^{th}$ spike, and $\phi_I(t)$ is the instantaneous phase of input $I$ at frequency $\omega$.

**4.3.1 Rhythmic inputs.**   Periodic pulse inputs were given by the expression

$$I_{PP}(t) = g_{PP}\Sigma_i \chi_{\{|t-t_i^*|<=w(s-1)/2s\}}(t) * \exp\left(-(st/w)^2\right), \qquad (2)$$

where $t_i^* = 2\pi\omega i$ for $i = 1,2,\ldots$ is the set of times at which pulses occur, $\omega$ is the frequency, $w = 1000d/\omega$ is the pulse width given the duty cycle $d \in (0,1)$, $*$ is the convolution operator, and $s$ determines how square the pulse is, with $s = 1$ being roughly normal and higher $s$ being more square. For our simulations, we took $d = 1/4$ and $s = 25$, and $\omega$ ranged over the set {0.25, 0.5, 1, 1.5, . . ., 22.5, 23}. Input pulses were normalized so that the total (integrated) input was 1 pA/s, and were then multiplied by a conductance varying from 0 to 4 in steps of 0.1.

For $I_{PP}$, the instantaneous phase $\phi_I(t)$ was obtained as the angle of the complex time series resulting from the convolution of $I_{PP}$ with a complex Morlet wavelet having the same frequency as the input and a length of 7 cycles.

**4.3.2 Quasi-rhythmic inputs.**   Variable-duration pulse trains were given by the expression

$$I_{VP}(t) = g_{VP}\Sigma_i \chi_{\{|t-t_i^*-o_i|<=w_i\frac{(s_i-1)}{2s_i}\}}(t) * \exp\left(-\left(\frac{s_i t}{w_i}\right)^2\right), \qquad (3)$$

where

$$t_i^* = \Sigma_{j=1}^i 1000/\omega_j,$$

the frequencies $\{\omega_i\}_1^n$ are chosen uniformly from $[f_{low}, f_{high}]$, the pulse width is given by $w_i = 1000d_i/\omega_i$, the duty cycles $\{d_i\}_1^n$ are chosen uniformly from $[d_{low}, d_{high}]$, the shape parameters $\{s_i\}_1^n$ are chosen uniformly from $[s_{low}, s_{high}]$, and the offsets $\{o_i\}_1^n$ are chosen uniformly from $[o_{low}, o_{high}]$. For our simulations, these parameters are given in Table 5.

**Table 5. Varied pulse input ($I_{VP}$) parameters (see *Methods: Phase-locking to rhythmic and quasi-rhythmic inputs: Inputs* for details).**

| Input Bandwidth (= $f_{high}$–$f_{low}$) | $f_{low}$ | $f_{high}$ | $d_{low}$ | $d_{high}$ | $s_{low}$ | $s_{high}$ | $o_{low}$ | $o_{high}$ |
|---|---|---|---|---|---|---|---|---|
| 1 | 6.5 | 7.5 | 0.25 | 0.3 | 10 | 40 | 0 | 0.05 |
| 1.65 | 6.175 | 7.825 | 0.2375 | 0.325 | 10 | 41 | 0 | 0.1 |
| 2.3 | 5.85 | 8.15 | 0.225 | 0.35 | 9 | 41 | 0 | 0.15 |
| ⋮ | ⋮ | ⋮ | ⋮ | ⋮ | ⋮ | ⋮ | ⋮ | ⋮ |
| 13.35 | 0.325 | 13.675 | 0.0125 | 0.775 | 1 | 50 | 0 | 1 |

Since $I_{VP}$ was composed of pulses and interpulse periods of varying duration, it was not "oscillation-like" enough to employ standard wavelet and Hilbert transforms to obtain accurate estimates of its instantaneous phase. Instead, the following procedure was used to obtain the instantaneous phase of $I_{VP}$. First, the times that $\chi_{VP}$ went from zero to greater than zero ($\{a_i\}_{i=1}^n$) and from greater than zero to zero ($\{b_i\}_{i=1}^n$) were obtained. Second, we specified the phase of $I_{VP}$ on these points via the function $\phi_I^0(t)$, a piecewise constant function satisfying

$$\frac{d}{dt}\phi_I^0(t) = \sum_{i=1}^n \left( \frac{3\pi}{2}\delta_{a_i}(t) + \frac{\pi}{2}\delta_{b_i}(t) \right),$$

where $\delta$ is the Dirac delta function. Finally, we determined $\phi_I(t)$ from $\phi_I^0(t)$ via linear interpolation, i.e. by setting $\phi_I(t)$ to be the piecewise linear (strictly increasing) function satisfying

$$\phi_I(0) = 0, \qquad \phi_I(a_i) = \phi_I^0(a_i), \qquad \phi_I(b_i) = \phi_I^0(b_i).$$

The resulting function $\phi_I(t)$ advances by $\pi/2$ over the support of each input pulse (the support is the interval of time over which the input pulse is nonzero), and advances by $3\pi/2$ over the time interval between the supports of consecutive pulses.

**4.3.3 Speech inputs.**   Speech inputs were comprised of 20 blindly selected sentences from the TIMIT corpus of read speech [63], which contains broadband recordings of 630 speakers of eight major dialects of American English, each reading ten phonetically rich sentences. The 16 kHz speech waveform file for each sentence was processed through a model of subcortical auditory processing [64], which decomposed the input into 128 channels containing information from distinct frequency bands, reproducing the cochlear filterbank, and applied a series of nonlinear filters reflecting the computations taking place in subcortical nuclei to each channel. We selected 16 of these channels—having center frequencies of 0.1, 0.13, 0.16, 0.21, 0.26, 0.33, 0.41 0.55, 0.65, 0.82, 1.04, 1.31, 1.65, 2.07, 2.61, and 3.29 kHz—for presentation to our computational models. We varied the multiplicative gain of the resulting waveforms from 0 to 2 in steps of 0.1 to obtain inputs at a variety of strengths. Speech onset occurred after one second of simulation.

Like varied pulse inputs, speech inputs were not "oscillation-like" enough to estimate their instantaneous phase using standard wavelet and Hilbert transforms. Thus, we used the following procedure to extract the instantaneous phase of $I_{speech}$. First, we calculated the power spectrum of the auditory cortical input channel derived from the speech waveform, using the Thompson multitaper method. Second, we identified peaks in the power spectrum that were at least 2 Hz apart, and used the 2nd, 3rd, and 4th largest peaks in the power spectrum to identify the frequencies of the main oscillatory modes in the $\theta$ frequency band (the largest peak in the power spectrum was in the $\delta$ frequency band for the sentences we used). Then, we convolved the auditory input with Morlet wavelets at these three frequencies and summed the resulting complex time series, to obtain a close approximation of the $\theta$-frequency oscillations in the input. Finally, we took the angle of this complex time series at each point in time to be the instantaneous phase of the input at that channel.

While the distribution of the (spike rate adjusted) *PLV* was not normal even after log transformation, the ANOVA is robust to violation of non-normality, so we compared *PLV* across models, sub-bands, gains, and sentences by running a 4-way ANOVA, with gain as a continuous variable. All effects were significant, and post-hoc tests for sub-bands were run to identify the optimal sub-band across models (S2 Fig). We then compared *PLV* values from simulations conducted with inputs from 1000 sentences at this gain and sub-band, by running a 2-way ANOVA with sentence and model as grouping variables; post-hoc model comparisons are shown in S2 Fig.

## 4.4 Speech segmentation

To determine whether the activity of our models could contribute to accurate speech segmentation, we used a sum-and-threshold method to derive putative syllabic boundaries from the activity of each model. We then compared these model-derived boundaries to syllable boundaries derived from the phonemic transcriptions of each sentence, and determined how frequently model-derived boundaries occurred for each phoneme class.

**4.4.1 Model-derived syllable boundaries.**   To determine model-derived syllable boundaries, we first divided the auditory frequency range into 8 sub-bands consisting of 16 (adjacent) channels each. For each sub-band and each model, the output from these 16 channels was used to drive the RS cells in 16 identical but unconnected versions of the model, with a multiplicative gain that varied from 0 to 2 in steps of 0.2. To approximate the effect these RS cells might have on a shared postsynaptic neuron, their time series of spiking activity, given by $\{s_i(t)\}_1^{16}$, were convolved with an exponential kernel having decay time $w_s/5$, summed over cells, and smoothed with a gaussian kernel with $\sigma = 25/4$ ms:

$$P(t) = \sum_{i=1}^{16} \left( s_i(t) * \exp\left(-\frac{5t}{w_s}\right) \right) * \frac{1}{\sigma\sqrt{2\pi}} \exp\left(-\frac{1}{2}\left(\frac{t}{\sigma}\right)^2\right).$$

The maximum of this "postsynaptic" time series during the second prior to speech input was then used to determine a threshold

$$p^* = r_{\text{thresh}}\max\{P(t)|t <= 1000\text{ms}\}$$

and the ordered set of times $\{m_i^*\}$ at which $P(t)$ crossed $p^*$ from below were extracted as candidate syllable boundaries. Starting with $i = 2$, any candidate boundary $m_i^*$ that followed the previous candidate boundary $m_{i-1}^*$ with a delay less than a refractory period of 25 ms was removed from $\{m_i^*\}$ to yield a set of model-derived syllable boundaries $m = \{m_i\}_1^{n_m}$.

**4.2.2 Transcription-derived syllable boundaries.**   Phoneme identity and boundaries have been labelled by phoneticians in every sentence of the TIMIT corpus. We used the Tsylb2 program [126] that automatically syllabifies phonetic transcriptions [127] to merge these sequences of phonemes into sequences of syllables according to English grammar rules, and thus determine the (transcription-derived) syllable boundary times $\{t_i^*\}_1^{n_s}$ for each sentence. The syllable midpoints were the set $t = \{t_i\}_1^{n_t}$ obtained by averaging successive pairs of syllable boundaries,

$$t_i = (t_i^* + t_{i+1}^*)/2, \quad i = 1, \ldots, n_s - 1 =: n_t.$$

**4.4.3 Comparing model- and phoneme-derived syllable boundaries.**   To compare the sets $m$ and $t$ for each sentence, we used a recursively-computed point-process metric [65]. This metric is defined by

$$d_{\text{VP},\tau}(m, t) = \min_{\{m=s^1,s^2,\ldots,s^l=t\}} \Sigma_1^l C(s^i, s^{i+1}),$$

where $\tau$ is a defining timescale, and $m$, $t$, and each $s^i = \{s_1^i, \ldots, s_{n_i}^i\}$ are series of boundary times, with $s^i$ and $s^{i+1}$ differing by at most one boundary (which can be altered, added, or removed). The "cost" of each "move" in the chain of (series of) boundary times $s^1, s^2, \ldots, s^1$ is

given by

$$
C(s^i, s^{i+1}) = \begin{cases} |s^i_l - s^{i+1}_m|/\tau, & \max_{\substack{j=1,\ldots,n_i \\ k=1,\ldots,n_{i+1}}} |s^i_j - s^{i+1}_k| < \tau, \\ 1, & \text{otherwise}. \end{cases}
$$

In other words, the cost of moving one boundary by a distance less than $\tau$ is less than 1, while the costs of shifting a boundary by $\tau$ or more, adding a boundary, and removing a boundary are all 1. It is helpful to note that

$$
\lim_{\tau \to \infty} d_{\text{VP},\tau}(m, t) = |n_m - n_t|, \qquad \lim_{\tau \to 0} d_{\text{VP},\tau}(m, t) = n_m + n_t.
$$

Since $d_{\text{VP},\tau}(m,t)$ as defined above scales with $\max(n_m, n_t)$, we normalized this distance by the number of moves that cost less than 1, and the n log-transform it, defining

$$
D_{\text{VP},\tau}(m, t) = \log \left( \frac{{}_{\text{VP},\tau}(m, t)}{\#\{\hat{s}^i | C(\hat{s}^{i-1}, \hat{s}^i) < 1\}} \right),
$$

where the sequence $\{m = \hat{s}^1, \ldots, \hat{s}^l = t\}$ realizes the minimum defining $D_{\text{VP},\tau}(m,t)$. Thus, $D_{\text{VP},\tau}(m,t) < 0$ if each boundary in $m$ corresponds to a distinct boundary in $t$ shifted by less than or equal to $\tau$, and all other things being equal, this normalized distance penalizes both missed and extra model-derived syllable boundaries. We used a timescale of $\tau = 50$ ms.

**4.4.4 Comparing segmentation across models.** To "optimize" the thresholding process for each model, we chose the pair of values from the sets $w_s = \{25, 30, \ldots, 75\}$ and $r_{\text{thresh}} = \{1/3, .4, .45, \ldots, .6, 2/3\}$ that minimized the minimum (over input channels and gains) of the mean of $D_{\text{VP},50}$ for 40 randomly chosen sentences. We then analyzed the distribution of $D_{\text{VP},50}$ at these model-specific "optimal" values of $w_s$ and $r_{\text{thresh}}$. The distribution of $D_{\text{VP},50}$ for each model was determined by the Kolmogorov-Smirnov test to be normal, so we compared $D_{\text{VP},50}$ across models, sub-bands, gains, and sentences by running a 4-way ANOVA. All effects were significant, and post-hoc tests for sub-bands and gains were run to identify the optimal gain and sub-band across models (S4 Fig). We then compared $D_{\text{VP},50}$ values from simulations with inputs at this gain and sub-band extracted from 1000 sentences. After again "optimizing" $w_s$ and $r_{\text{thresh}}$ for each model, we ran a 2-way ANOVA with sentence and model as grouping variables; post-hoc tests are shown in S4 Fig.

**4.4.5 Phoneme distributions of model boundaries.** To determine the phoneme distributions of model boundaries, we used the phonemic transcriptions from the TIMIT corpus. The time of each model-derived boundary was compared to the set of onset and offset times of phonemes to determine the identity of the phoneme at boundary occurrence. For each simulation, we constructed a histogram over all phonemes in the TIMIT corpus; we then combined the histograms across simulations, and multiplied them by a matrix whose rows were indicator functions for 7 different phoneme classes—stops, affricates, fricatives, nasals, semivowels and glides, vowels, and other, a category which included pauses. We performed the same procedure for the set of mid-syllable times for each sentence we used in the corpus to obtain the phoneme distribution at mid-syllable.

## 4.5 Spike-triggered input pulses

To explore the buildup of outward current and delay of subsequent spiking induced by strong forcing, we probed each model with a single spike-triggered pulse. These pulses were triggered by the first spike after a transient interval of 2000 ms, had a pulse duration of 50 ms, and had a

form given by the summand in Eq (2) with $w = 50$ and $s = 25$ ($i$ was 1 and $t_i$ was the time of the triggering spike).

## Supporting information

**S1 Fig. Dependence of one-to-one phase locking on inhibitory conductance.** We multiplied the conductances $g_{\mathrm{m}}$ and $g_{\mathrm{inh}}$ in model MIS by factors of $\frac{1}{3}$, $\frac{1}{2}$, $\frac{3}{4}$, 1, and $\frac{5}{4}$, and then computed plots of PLV for different input frequencies and strengths, as in Fig 3. The bright yellow band in each figure, representing the region of one-to-one phase-locking, depends on the size of $g_{\mathrm{m}}$ and $g_{\mathrm{inh}}$; both increase from left to right.
(EPS)

**S2 Fig. Statistical tests of PLV.** PLV depended linearly on input gain (left), as shown by a plot of the joint density of input gain and PLV, along with the regression line of PLV onto input gain (white, $p < 10^{-10}$). In an ANOVA with gain treated as a continuous regressor, the group effect for channels was highly significant (middle, $p < 10^{-10}$); lines connect channels that are *not* significantly different in post-hoc tests at level $\alpha = .05$. In a separate ANOVA for results from simulations with input from 1000 sentences at only the optimal gain and channel, post-hoc tests showed significant differences between all models at level $\alpha = .05$.
(EPS)

**S3 Fig. Segmentation performance depends on threshold.** False-color plots show the mean $D_{\mathrm{VP},50}$ for different auditory sub-bands (x-axis) as well as varying input strengths (y-axis) for all six models, with model-derived boundaries determined by the parameters $w_s = 75$ and $r_{\mathrm{thresh}} = 1/3$ (left), $r_{\mathrm{thresh}} = 0.45$ (middle left), $r_{\mathrm{thresh}} = 0.55$ (middle right), and $r_{\mathrm{thresh}} = 2/3$. The model exhibiting the best segmentation performance shifts with the value of $r_{\mathrm{thresh}}$.
(EPS)

**S4 Fig. Statistical tests of $D_{\mathrm{VP},50}$.** In an ANOVA treating input gain (left), sub-band center frequency (middle), and model as categorical variables, all effects were highly significant ($p < 10^{-10}$). Lines connect channels that are *not* significantly different in post-hoc tests at level $\alpha = .05$. In a separate ANOVA for results from simulations with input from 1000 sentences at only the optimal gain and channel, post-hoc tests clustered the models in four groups at level $\alpha = .05$ (right).
(EPS)

**S5 Fig. Dynamics of inhibitory currents in models MIS and MI.** Plots of the pre-spike gating variables in models MS, MIS, and MI. Top row, plotting the second difference in m-current activation level of against its first difference reveals that pre-spike activation levels are clustered along a single branch of the oscillator's trajectory. Middle row, plots of the relationships between the pre-spike activation levels of $I_{\mathrm{inh}}$, $I_{\mathrm{m}}$, and $I_{\mathrm{K_{SS}}}$ in model MIS, revealing a dependence on the phase of oscillations in m-current activation. Bottom, plots of the relationships between the pre-spike activation levels of $I_{\mathrm{inh}}$ and $I_{\mathrm{m}}$ in model MI, again revealing a dependence on the phase of oscillations in m-current activation. (For all plots, light gray curves represent trajectories with an input pulse; dark gray curves represent trajectories without an input pulse).
(EPS)

**S6 Fig. Varying tonic input to model MS.** We altered the tonic input strength $g_{\mathrm{app}}$ to model MS, and gave periodic pulse inputs of strength $g_{\mathrm{PP}} = 1$ at varying frequencies. For lower levels of tonic input, phase-locking is closer to one-to-one for low frequency inputs, but many high frequency input cycles are "missed"; for higher levels of tonic input, phase-locking is one-to-

one for high frequency inputs, but many-to-one for low frequency inputs.
(EPS)

## Acknowledgments

We thank Oded Ghitza and Laura Dilley for many useful discussions.

## Author Contributions

**Conceptualization:** Benjamin R. Pittman-Polletta, Yangyang Wang, David A. Stanley, Charles E. Schroeder, Miles A. Whittington, Nancy J. Kopell.

**Formal analysis:** Benjamin R. Pittman-Polletta.

**Funding acquisition:** Charles E. Schroeder, Nancy J. Kopell.

**Investigation:** Benjamin R. Pittman-Polletta, Yangyang Wang, David A. Stanley, Charles E. Schroeder, Miles A. Whittington, Nancy J. Kopell.

**Software:** Benjamin R. Pittman-Polletta, David A. Stanley.

**Visualization:** Benjamin R. Pittman-Polletta.

**Writing – original draft:** Benjamin R. Pittman-Polletta.

**Writing – review & editing:** Benjamin R. Pittman-Polletta, Yangyang Wang, David A. Stanley, Charles E. Schroeder, Miles A. Whittington, Nancy J. Kopell.

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
