## [Decision Letter · Decision Letter 0]

27 Mar 2020

Dear Dr. Pittman-Polletta,

Thank you very much for submitting your manuscript "Differential contributions of synaptic and intrinsic inhibitory currents to speech segmentation via flexible phase-locking in neural oscillators" for consideration at PLOS Computational Biology.

As with all papers reviewed by the journal, your manuscript was reviewed by members of the editorial board and by several independent reviewers. In light of the reviews (below this email), we would like to invite the resubmission of a significantly-revised version that takes into account the reviewers' comments.

We cannot make any decision about publication until we have seen the revised manuscript and your response to the reviewers' comments. Your revised manuscript is also likely to be sent to reviewers for further evaluation.

Sincerely,

Boris S. Gutkin

Associate Editor

PLOS Computational Biology

Kim Blackwell

Deputy Editor

PLOS Computational Biology

Reviewer's Responses to Questions

**Comments to the Authors:**

Reviewer #1: The manuscript describes a modeling work that explores the influence of different inhibitory currents on the phase locking properties of theta oscillations. The manuscript is well written, structured, and represents an interesting study, providing useful novel notions for future modeling in the domain of speech processing.

I do not have strong criticisms but a few points could perhaps be improved/specified

Introduction:

What is the necessity of having a model phase-locked to rhythms slower than its intrinsic frequency? Since, as the authors suggest, there is no problem for the majority of models to lock to faster frequencies, a flexible oscillator could more easily be achieved by setting the intrinsic frequency at the lowest bound of the theta range.

L25: I would not lump together beta/gamma (15-60 Hz) as it covers a range of diverse possible functions.

Results:

If, as the authors hypothesize, the model provides mechanism for flexible theta tracking, then the model should exhibit degradation of phase-locking at frequencies close to the upper bound of theta range. However, this is not clearly demonstrated in the results. Several candidate models still have high PLVs above 15 Hz, given a strong enough input strength.

Discussion:

L390: distinguish “additive” and “synergistic” more specifically?

L403: the claim “neurons in deep cortical layers are likely to exhibit all three currents” need reference.

L455: ref missing?

L481 – 485: I think the authors mixed some numbers (average duration of a spoken syllable), or the reference (in the cited paper, they compress speech by x3 (40ms) chunks and inset silent gap from 0-120ms, the famous U-shape). The optimal performance there occurs when 40 ms speech chunk is followed by 80 ms silence chunk -> resulting in around 6Hz (120ms). Thus, they are right, if the average syllable duration is 333ms, then x3 compression would put the syllabic rate above 9Hz, and inserting silence according to U-shape (666ms?) would put it below 9Hz optimal rate. I am just having a problem, about where they take 333ms as average syllable duration. In the case of 200ms (5Hz), as reported in Greenberg (1999), and other studies, 3x factor would lead to 15Hz, out of theta range. In any case, I found this paragraph hard to follow, and rephrasing it would be desirable.

L501: it would be helpful if the authors can suggest where these MS neurons are located in the auditory cortex.

Reviewer #2: The study by Benjamin Pittman-Polletta and colleagues addresses an interesting scientific question: how can neural oscillations be flexible enough to lock to quasi-rhythmic sensory signal such as the syllabic rate in speech? However, there are in my opinion several major shortcomings that severely limit the impact of the results (listed below). In the end, the study is stuck halfway between two possible outcomes: on one hand it does not a theoretical account of how neural oscillator can reliably lock down to an external input whose internal frequency fluctuates (although it features interesting phenomenological observations); on the other hand, there is no evidence that the novel model detects syllable boundaries better than existing models.

1- From what I understood that the strategy to avoid missing one pulse is to accumulate low time scales in neural dynamics in the oscillator. But then what is the point in using an oscillator in the first place, and not trivially a neuron that only reaches spiking threshold when a pulse is provided?

2- Syllable boundaries do not correspond to the high-energy vocalic portion of the syllable, but just the opposite: the low-energy portions corresponding to closure of the vocal tract. The vowel is the center (nucleus) of the syllable. Cutting a word such as “Badu” at high-energy portions would lead to 3 “syllables”: “ba”, “adu” and “u”. Clearly not the most conventional definition of syllables…

3 – It is not clear whether the proposed mechanisms allow better syllable detection than previous oscillator-based models of syllable detection (Hyafil et al, 2015; Räsänen et al., 2018). In particular, speech signal is far more complex than the input used here, with a spectrum likely dominated by 1/f component, so we would need to see how the proposed models behave in response to such signal. Second, it is not clear at all what level of phase-locking is required to accurately detect syllable boundaries, so it would really help to test actual syllable boundary detection, e.g. using the methodology developed in one of the above-mentioned studies.

4- The manuscript lacks clarity. A lot of things could/should be improved: it is very hard to follow the rationale for all the different mechanisms (the architecture seems completely arbitrary, until we get some intuition in Figures 5-6), as well as the specifics of all 5 models; the Methods section is very difficult to follow as it is, placed before Results; Introduction section is too long; some elements are explained twice; some figures panels are not commented in main text (e.g. FI curves), some figure labels and panel labels are missing (e.g. Fig 2D), etc.

MINOR POINTS

- How do SOM neuron respond?

- Is the architecture of SOM neurons taken from any existing reference?

- what is chi(t) line 119?

- a plot/inset of periodic pulses and quasi-period pulses would help

- why is baseline frequencies not lower for models with more inhibitory currents?

- The Hilbert transform is a more principled method for extracting the phase of the input signals than the one used here

- Figure 1: why are there 2 FI plot for each curve?

- why use ‘outward current’ and not ‘inhibitory current’ consistently?

- the last sentence of the abstract mentions something about the neural oscillator allowing “reliable time-keeping”, but I found no reference to this function in the manuscript.

**Have all data underlying the figures and results presented in the manuscript been provided?**

Reviewer #1: Yes

Reviewer #2: No: will be made available upon publication

PLOS authors have the option to publish the peer review history of their article (what does this mean?). If published, this will include your full peer review and any attached files.

Reviewer #1: No

Reviewer #2: No
---

## [Decision Letter · Decision Letter 1]

1 Oct 2020

Dear Dr. Pittman-Polletta,

Thank you very much for submitting your manuscript "Differential contributions of synaptic and intrinsic inhibitory currents to speech segmentation via flexible phase-locking in neural oscillators" for consideration at PLOS Computational Biology. As with all papers reviewed by the journal, your manuscript was reviewed by members of the editorial board and by several independent reviewers. The reviewers appreciated the attention to an important topic. Based on the reviews, we are likely to accept this manuscript for publication, providing that you modify the manuscript according to the review recommendations.

Sincerely,

Boris S. Gutkin

Associate Editor

PLOS Computational Biology

Kim Blackwell

Deputy Editor

PLOS Computational Biology

[LINK]

Reviewer's Responses to Questions

**Comments to the Authors:**

Reviewer #1: The previous comments have been properly addressed, therefore I only have minor comments here.

L142: typo, “rage” range

L201: “phase locking of model MS… by a lack of spiking when there is no speech input”. This also seems to be the case for other types. How was this conclusion drawn? It would be helpful to have a quantitative metric, e.g. correlation between the speech envelope and instantaneous firing rate of the neuron.

Fig.6: need color bar for the phase locking values on the right-side plot

L456-457: “may time relevant calculations…” please correct.

Reviewer #2: The revised manuscript constitutes in my opinion a major improvement in comparison to the original submission. It has gained a lot in clarity, concepts are much better articulated, and my major concerns have been addressed – or at least clarified. I am still quite uncomfortable with using the term “speech segmentation” in the title though, since really all the work is about measuring phase-locking and authors have not checked whether speech signal would be segmented in any meaningful manner. I also strongly suggest to openly report this in the Discussion as a limitation of the study: flexibility is very nice but it remains to be shown that such model outperforms previous models of speech segmentation. Here is a list of minor comments below:

- Label for Y-axis is missing on figure 2B bottom

- Legend figure 2 D: calibration means the scale of the horizontal and vertical bar?

- L256: “each spike suggests that the two gating variables are negatively linearly related” -> specifiy: “at spike times”

- Figure 8 is messy, legend says we should see no input pulse but I cannot see any ‘x’, the regression line is also hard to see, consider using smaller symbols or points. Missing parenthesis in capture.

- Sentence L301-307 is very long and difficult to understand, consider simplifying.

Formulation of sentence L423-427 is awkward

- Chi(t) is still not explained when it is introduced L557.

- Please explain the rationale for the change in applied current after 500 ms L557.

- Is the negative sign correct for Iext in equation L561?

- L564: “conductance values for all six models that will be introduced in Results:” -> correct future tense

**Have all data underlying the figures and results presented in the manuscript been provided?**

Reviewer #1: Yes

Reviewer #2: **No: **

PLOS authors have the option to publish the peer review history of their article (what does this mean?). If published, this will include your full peer review and any attached files.

Reviewer #1: No

Reviewer #2: No
---

## [Editor Report · Decision Letter 2]

5 Feb 2021

Dear Dr. Pittman-Polletta,

We are pleased to inform you that your manuscript 'Differential contributions of synaptic and intrinsic inhibitory currents to speech segmentation via flexible phase-locking in neural oscillators' has been provisionally accepted for publication in PLOS Computational Biology.

Best regards,

Boris S. Gutkin

Associate Editor

PLOS Computational Biology

Kim Blackwell

Deputy Editor

PLOS Computational Biology

---

## [Editor Report · Acceptance letter]

9 Apr 2021

PCOMPBIOL-D-20-00190R2 

Differential contributions of synaptic and intrinsic inhibitory currents to speech segmentation via flexible phase-locking in neural oscillators

Dear Dr Pittman-Polletta,

I am pleased to inform you that your manuscript has been formally accepted for publication in PLOS Computational Biology. Your manuscript is now with our production department and you will be notified of the publication date in due course.

With kind regards,

Andrea Szabo
